# FEDERATED CONTRASTIVE GFLOWNETS

## ABSTRACT

Generative flow networks (GFlowNets) are powerful samplers for distributions supported in spaces of compositional objects (e.g., sequences and graphs), with applications ranging from the design of biological sequences to causal discovery. However, there are no principled approaches to deal with GFlowNets in federated settings, where the target distribution results from a combination of (possibly sensitive) rewards from different parties. To fill this gap, we propose *federated contrastive GFlowNet* (FC-GFlowNet), a divide-and-conquer framework for federated learning of GFlowNets, requiring a single communication step. First, each client learns a GFlowNet locally to sample proportionally to their reward. Then, the server gathers the local policy networks and aggregates them to enforce *federated balance* (FB), which provably ensures the correctness of FC-GFlowNet. Additionally, our theoretical analysis builds on the idea of *contrastive balance*, that imposes necessary and sufficient conditions for the correctness of general (non-federated) GFlowNets. We empirically attest the performance of FC-GFlowNets in four settings, including grid-world, sequence, multiset generation, and Bayesian phylogenetic inference. Experiments also suggest that, in some cases, enforcing the contrastive balance can accelerate the training of conventional GFlowNets.

## 1 INTRODUCTION

Generative flow networks (GFlowNets, Bengio et al., 2021; 2023) is a family of reward-driven generative models for compositional objects (e.g., sequences or graphs). While GFlowNets were originally designed to increase diversity of candidates in active learning settings, they have found applications in a variety of domains, such as causal discovery (Deleu et al., 2022; 2023; da Silva et al., 2023), combinatorial optimization (Zhang et al., 2023b), design of biological sequences (Jain et al., 2022), drug discovery (Bengio et al., 2021), and multiobjective optimization (Jain et al., 2023).

In essence, GFlowNets cast the problem of sampling from an unnormalized distribution/reward as a network flow problem (Bazaraa et al., 2004). Starting from an initial state, GFlowNets iteratively draw actions according to a (forward) policy which, in turn, increments the state — eventually creating valid samples. This process can be interpreted as spreading the total mass of the distribution (flow at the source) through trajectories that lead to elements in the target distribution's support (sink nodes). In practice, GFlowNets are learned by enforcing balance conditions (Malkin et al., 2022; Pan et al., 2023a) that ensure the correctness of the sampling distribution.

There are many applications where the target reward decomposes as a product of (local) reward functions held by different clients/parties. For instance, this appears in distributed Bayesian inference (Neiswanger et al., 2014; El Mekkaoui et al., 2021), where the reward involves a likelihood function, which is typically log-additive on data shards — the posterior (i.e., target reward) is proportional to a product of the prior and the local likelihoods defined on each client's data, each of which can be seen as a local reward. Another relevant application is multi-objective active learning (Daulton et al., 2021), where we wish to obtain a collection of diverse designs with high values under a pool of utility functions. For instance, in drug discovery, a key challenge is finding molecules that satisfy multiple constraints (e.g., affinity, solubility, safety). In a setting where different experts work independently to obtain property-specific models using GFlowNets, this challenge can be cast as sampling from the product of these local GFlowNets (Garipov et al., 2023). Furthermore, in both cases, when the local rewards are of a sensitive nature, clients might be reluctant to share them directly (e.g., as it may incur sharing data). While this falls into the realm of *Federated Learning* (FL, McMahan et al., 2017), so far, there are no works on FL of GFlowNets.

**Reviewer:** wWFS, Npeg, rsSg

This paper extends the theory of GFlowNets to develop a simple *divide-and-conquer* algorithm for federated learning of GFlowNets, in which clients learn locally GFlowNets to sample from their individual rewards and, subsequently, send them to the server for aggregation. Importantly, this procedure requires a single round of communication between the client and server sides. More specifically, we derive the *federated balance* condition to ensure the GFlowNet samples from the product of (the marginal distribution over terminal states induced by) the local GFlowNets. Furthermore, on our path towards federated balance, we derive a novel balance criterion to train non-federated GFlowNets, the *contrastive balance* (CB) condition. Informally, the CB encapsulates the idea i) that we can 'estimate' the normalizing constant of a reward given a terminal trajectory using the trajectory balance condition (Malkin et al., 2022), and ii) if a GFlowNet is perfectly trained, the estimate should be independent of the trajectory used. Enforcing the CB induces a loss that, in expectation, is equivalent to the variance of an estimator of the log-partition, whose minimization was first proposed by Zhang et al. (2023a).

> **Reviewer:** Xj4h

Our experiments on four different federated tasks show that our method, *federated contrastive GFlowNet* (FC-GFlowNet), can accurately sample from the combined rewards without direct access to each client's reward — only to their locally trained GFlowNets. For conventional (non-federated) settings, we also show that using the contrastive balance as a training criterion leads to better convergence when the intersection of the GFlowNet's sets of terminal and intermediate states is void.

In summary, our **contributions** are:

1. We propose the first algorithm for federated learning of GFlowNets. Our method incurs a divide-and-conquer framework, in which clients learn GFlowNets based on their private rewards and send their policies to a server for aggregation. We provide a theory that guarantees its correctness, and also analyze its robustness to errors in the estimation of local GFlowNets;

2. We present the contrastive balance condition, which can be used to train general GFlowNets. We show it is a sufficient and necessary condition for sampling proportionally to a reward and analyze its connection to variational inference (VI);

3. We substantiate our methodological contributions with experiments on four different tasks. Notably, our empirical results $i$) demonstrate the accuracy of our federated framework for GFlowNets; $ii$) show that, in some cases, using the contrastive balance as a training criterion leads to faster convergence rates compared to using trajectory and detailed balances; iii) illustrate the potential of GFlowNets in a novel application: Bayesian phylogenetic inference.

## 2 PRELIMINARIES

**Notation.** We represent a *directed acyclic graph* (DAG) over nodes $V$ and with adjacency matrix $A \in \{1, 0\}^{|V| \times |V|}$ as $G = (V, A)$. A *forward policy* over $V$ in $G$ is a function $p: V \times V \to \mathbb{R}_+$ such that (i) $p(v, \cdot)$ is a probability measure over $V$ for every $v \in V$ and (ii) $p(v, w) > 0$ if and only if $A_{vw} = 1$; we alternatively write $p(v \to w)$ to represent $p(v, w)$. Characteristically, a transition kernel $p$ induces a conditional probability measure over the space of trajectories in $G$: if $\tau = (v_1 \to v_2 \to \cdots \to v_n)$ is a trajectory of length $n$ in $G$, then $p(\tau | v_1) = \prod_{1 \leq i \leq n-1} p(v_{i+1} | v_i)$. A *backward policy* in $G$ is a forward policy on the transpose graph $G^{\mathsf{T}} = (V, A^{\mathsf{T}})$.

**Generative flow networks.** GFlowNets are a family of amortized variational algorithms trained to sample from an unnormalized distribution over discrete and compositional objects. More specifically, let $R: \mathcal{X} \to \mathbb{R}_+$ be an unnormalized distribution over a finite space $\mathcal{X}$. We call $R$ a *reward* due to terminological inheritance from the reinforcement learning literature. Define a finite set $\mathcal{S}$ and a variable $s_o$. Then, let $G$ be a weakly connected DAG with nodes $V = \{s_o\} \cup \{s_f\} \cup \mathcal{S} \cup \mathcal{X}$ such that (i) there are no incoming edges to $s_o$, (ii) there are no outgoing edges exiting $s_f$ and (iii) there is an edge from each $x \in \mathcal{X}$ to $s_f$. We call the elements of $V$ *states* and refer to $\mathcal{X}$ as the set of *terminal states*; $s_o$ is called the *initial state* and $s_f$ is an absorbing state designating the end of a trajectory. We denote by $\mathcal{T}$ the space of trajectories in $G$ starting at $s_o$ and ending at $s_f$. Illustratively, $\mathcal{X}$ could be the space of biological sequences of length 32; $\mathcal{S}$, the space of sequences of lengths up to 32; and $s_o$, an empty sequence. The training objective of a GFlowNet is to learn a forward policy $p_F$ over $G$ such that the marginal distribution $p_T$ over $\mathcal{X}$ satisfies

$$p_T(x) := \sum_{\tau : \tau \text{ leads to } x} p_F(\tau | s_o) \propto R(x). \tag{1}$$

We usually parameterize $p_F$ as a neural network and select one among the diversely proposed training criteria to estimate its parameters, which we denote by $\phi_F$. These criteria typically enforce a balance condition on the Markovian process defined by $p_F$ that provably imposes the desired property on the marginal distribution in Equation 1. For example, the *trajectory balance* (TB) criterion introduces a parameterization of the target distribution's partition function $Z_{\phi_Z}$ and of a backward policy $p_B(\cdot, \cdot; \phi_B)$ with parameters $\phi_Z$ and $\phi_B$, respectively, and is defined as

$$\mathcal{L}_{TB}(\tau, \phi_F, \phi_B, \phi_Z) = \left( \log Z_{\phi_Z} - \log R(x) + \sum_{s \to s' \in \tau} \log \frac{p_F(s, s'; \phi_F)}{p_B(s', s; \phi_B)} \right)^2. \tag{2}$$

Minimizing Equation 2 enforces the TB condition: $p_F(\tau; \phi_F) = Z_{\phi_Z}^{-1} R(x) \prod p_B(s', s; \phi_B)$, which implies Equation 1 if valid for all $\tau \in \mathcal{T}$. This is the most widely used training scheme for GFlowNets. In practice, some works set $p_B$ as a uniform distribution to avoid learning $\phi_B$, as suggested by Malkin et al. (2022).

Another popular approach for training GFlowNets uses the notion of *detailed balance* (DB). Here, we want to find forward and backward policies and *state flows* $F$ (with parameters $\phi_S$) that satisfy the DB condition: $F(s; \phi_S) p_F(s, s'; \phi_F) = F(s'; \phi_S) p_B(s', s; \phi_B)$ if $s$ is an non-terminal state and $F(s; \phi_S) p_F(s_f | s; \phi_F) = R(s)$ otherwise. Again, satisfying the DB condition for all edges in $G$ entails Equation 1. Naturally, this condition leads to a transition-decomposable loss

$$\mathcal{L}_{DB}(s, s', \phi_F, \phi_B, \phi_S) = \begin{cases} \left( \log \frac{p_F(s, s'; \phi_F)}{p_B(s', s; \phi_B)} + \log \frac{F(s; \phi_S)}{F(s'; \phi_S)} \right)^2 & \text{if } s' \neq s_f, \\ \left( \log \frac{F(s; \phi_S) p_F(s_f | s; \phi_F)}{R(s)} \right)^2 & \text{otherwise.} \end{cases} \tag{3}$$

In recent work, Pan et al. (2023a) proposed a novel residual parameterization of the state flows that achieved promising results in terms of speeding up the training convergence of GFlowNets. More specifically, the authors assumed the existence of a function $\mathcal{E} \colon \mathcal{S} \to \mathbb{R}$ such that (i) $\mathcal{E}(s_o) = 0$ and (ii) $\mathcal{E}(x) = -\log R(x)$ for each terminal state $x \in \mathcal{X}$ and reparameterized the state flows as $\log F(s, \phi_S) = -\mathcal{E}(s) + \log \tilde{F}(s, \phi_S)$. This new training scheme was named *forward looking (FL) GFlowNets* due to the inclusion of partially computed rewards in non-terminal transitions.

To learn GFlowNet parameters, we need to average $\mathcal{L}_{DB}$ or $\mathcal{L}_{TB}$ over some exploratory policy $\pi$ fully supported in $\mathcal{T}$. In practice, $\pi$ is typically a $\epsilon$-mixture between $p_F$ and a uniform forward policy, $(1 - \epsilon) \cdot p_F + \epsilon \cdot u_F$, or a tempered version of $p_F$. We use the former definition for $\pi$ in this work. We review alternative training schemes for GFlowNets in the supplementary material.

**Problem statement.** We are interested in the federated setting where there is a set of clients $n = 1, \ldots, N$, each with reward function $R_n$, and we want to learn a GFlowNet to sample proportionally to a global reward function $R$ defined as a product of the local rewards $R_1, \ldots, R_N$ — under the restriction that clients are not willing to openly disclose their rewards. This might be the case when, e.g., $R$ is an unnormalized Bayesian posterior or when it encodes a multi-objective criterion. While we focus on sampling from $R(x) := \prod_{n=1}^{N} R_n(x)$ in the main paper, our supplementary material also provides extensions of our theoretical results to exponentially weighted rewards of the form $R(x) := \prod_{n=1}^{N} R_n(x)^{w_n}$ with $w_1, \ldots, w_N > 0$. For related works, see Appendix D.

> **Reviewer:** wWFS

**Overview of the solution.** To circumvent the restrictions imposed by the problem statement, we propose a divide-and-conquer scheme. First, each client trains their own GFlowNet to sample proportionally to their local reward. Next, each client $n$ sends the forward and backward policies $p_F^{(n)}$ and $p_B^{(n)}$ to a centralizing server. Then, the server estimates the policies $(p_F, p_B)$ of a novel GFlowNet that approximately samples from $R$ solely based on the local policies $\{(p_F^{(n)}, p_B^{(n)})\}_{n=1}^{N}$. More specifically, the server learns a GFlowNet whose marginal distribution $p_T$ over terminal states is proportional to the product of those from the clients $p_T^{(1)}, \ldots, p_T^{(N)}$.

## 3 METHOD

This section derives a provably correct framework for federated GFlowNets (Section 3.1). Towards this end, we introduce the *contrastive balance condition*, a new balance condition that requires mini-

mal parameterization. Additionally, Section 3.2 further explores this condition as a general objective for training conventional GFlowNets, which stems naturally from our theoretical developments.

## 3.1 FEDERATED GFLOWNETS

Recall that we are interested in sampling proportionally to a product of the rewards from users $n = 1, \ldots, N$, i.e., we want to sample from $R(x) := \prod_{n=1}^{N} R_n(x)$. However, users might be reluctant to openly disclose $R_n$ due to its possibly sensitive nature. Therefore, we are prevented from centralizing $R_1, \ldots, R_N$ and directly training a network to sample from their product. To circumvent this limitation, we propose a simple divide-and-conquer strategy. First, each user $n$ trains independently a GFlowNet with forward/backward policies $p_F^{(n)}$ and $p_B^{(n)}$ to sample from $R_n$. Then, the users send their policies to a server for a single aggregation step, in which the server combines the local GFlowNets into a new one, with forward/backward policies $p_F$ and $p_B$, that induces a distribution $p_T$ over final states $x \in \mathcal{X}$ proportional to $R$. Toward this end, Theorem 1 delineates a necessary and sufficient condition that guarantees the correctness of the aggregation phase, which we call *Federated balance condition*. It is worth mentioning that Theorem 1 builds directly on Lemma 1. However, we postpone discussing the latter to Section 3.2.

**Theorem 1** (Federated balance condition). *Let* $\left(p_F^{(1)}, p_F^{(1)}\right), \ldots, \left(p_F^{(N)}, p_F^{(N)}\right) : V^2 \to \mathbb{R}^+$ *be pairs of forward and backward policies from $N$ GFlowNets sampling respectively proportionally to* $R_1, \ldots, R_N : \mathcal{X} \to \mathbb{R}^+$. *Then, another GFlowNet with forward and backward policies $p_F, p_B \in V^2 \to \mathbb{R}^+$ samples proportionally to $R(x) := \prod_{n=1}^{N} R(x)$ if and only if the following condition holds for all terminal trajectories $\tau, \tau' \in \mathcal{T}$:*

$$\prod_{1 \leq i \leq N} \frac{\left(\prod_{s \to s' \in \tau} \frac{p_F^{(i)}(s,s')}{p_B^{(i)}(s',s)}\right)}{\left(\prod_{s \to s' \in \tau'} \frac{p_F^{(i)}(s,s')}{p_B^{(i)}(s',s)}\right)} = \frac{\left(\prod_{s \to s' \in \tau} \frac{p_F(s,s')}{p_B(s',s)}\right)}{\left(\prod_{s \to s' \in \tau'} \frac{p_F(s,s')}{p_B(s',s)}\right)}. \tag{4}$$

Based on Theorem 1, we can naturally derive a loss function that enforces Equation 4 and can be used to combine the locally trained GFlowNets. To guarantee the minimum of our loss achieves federated balance, it suffices to integrate Equation 4 against a distribution that attributes non-zero mass to every element in $\mathcal{T}^2$. It is important to note the federated balance loss is agnostic to which loss was used to learn the local GFlowNets, as it only requires their transition functions.

**Corollary 1** (Federated balance loss). *Let $p_F^{(i)}$ and $p_B^{(i)}$ be forward and backward transition functions such that $p_T^{(i)}(x) \propto R_i(x)$ for arbitrary reward functions $R_i$ over terminal states $x \in \mathcal{X}$. Also, let $\nu : \mathcal{T}^2 \to \mathbb{R}^+$ be some full-support probability distribution over pairs of terminal trajectories. Moreover, assume that $p_F(\cdot, \cdot; \phi_F)$ and $p_B(\cdot, \cdot; \phi_B)$ denote the forward/backward policies of a GFlowNet, parameterized by $\phi_F$ and $\phi_B$. The following statements are equivalent:*

1. *$p_T(x; \phi_F) \propto \prod_i R_i(x)$ for all $x \in \mathcal{X}$;*

2. *$\mathbb{E}_{(\tau, \tau') \sim \nu} \left[ \mathcal{L}_{Fed}(\tau, \tau', \phi_F, \phi_B) \right] = 0$ where for trajectories $\tau \rightsquigarrow x$ and $\tau \rightsquigarrow x'$*

$$\mathcal{L}_{Fed}(\tau, \tau', \phi_F, \phi_B) = \left( \log \frac{p_F(\tau; \phi_F) p_B(\tau'|x'; \phi_B)}{p_B(\tau|x; \phi_B) p_F(\tau'; \phi_F)} + \sum_{1 \leq i \leq N} \log \frac{p_F(\tau; \phi_F) p_B(\tau'|x'; \phi_B)}{p_B(\tau|x; \phi_B) p_F(\tau'; \phi_F)} \right)^2. \tag{5}$$

**Remark 1** (Imperfect local inference). In practice, the local balance conditions often cannot be satisfied by the local GFlowNets and the distributions $p_T^{(1)}, \ldots, p_T^{(N)}$ over terminal states are not proportional the the rewards $R_1, \ldots, R_N$. In this case, federated balance implies the aggregated model samples proportionally to

$$\mathbb{E}_{\tau \sim p_B(\cdot|x)} \left[ \prod_{1 \leq i \leq N} \frac{p_F^{(i)}(\tau)}{p_B^{(i)}(\tau|x)} \right]. \tag{6}$$

Figure 1: **FC-GFlowNet samples proportionally to a pool of locally trained GFlowNets.** If a client correctly trains their local model (green) and another client trains theirs incorrectly (red), the distribution inferred by FC-GFlowNet (mid-right) differs from the target product distribution (right).

Interestingly, the value of Equation 6 equals the expectation of a non-deterministic random variable only if the local balance conditions are not satisfied. Otherwise, the ratio $p_F^{(i)}(\tau)/p_B^{(i)}(\tau|x)$ equals $R(x)$, a constant wrt $\tau$ conditioned on $\tau$ having $x$ as its final state. Furthermore, Equation 6 also allows us to assess the probability mass function the global GFlowNet is truly drawing samples from.

As mentioned in Remark 1, in practice, the local GFlowNets may not be balanced with respect to their rewards, incurring errors that propagate to our aggregated model. In this context, Theorem 2 quantifies the extent to which these local errors impact the overall result.

**Theorem 2** (Influence of local failures). *Let $\pi_n := R_n/Z_n$ and $p_F^{(n)}$ and $p_B^{(n)}$ be the forward and backward policies of the $n$-th client. We use $\tau \rightsquigarrow x$ to indicate that $\tau \in \mathcal{T}$ is finished by $x \to s_f$. Suppose that the local balance conditions are lower- and upper-bounded $\forall\, n = 1, \dots, N$ as per*

$$1 - \alpha_n \leq \min_{x \in \mathcal{X}, \tau \rightsquigarrow x} \frac{p_F^{(n)}(\tau)}{p_B^{(n)}(\tau|x)\pi_n(x)} \leq \max_{x \in \mathcal{X}, \tau \rightsquigarrow x} \frac{p_F^{(n)}(\tau)}{p_B^{(n)}(\tau|x)\pi_n(x)} \leq 1 + \beta_n \qquad (7)$$

*where $\alpha_n \in (0,1)$ and $\beta_n > 0$. The Jeffrey divergence $\mathcal{D}_J$ between the global model $\hat{\pi}(x)$ that fulfills the federated balance condition in Equation 4 and $\pi(x) \propto \prod_{n=1}^{N} \pi_n(x)$ then satisfies*

$$\mathcal{D}_J(\pi, \hat{\pi}) \leq \sum_{n=1}^{N} \log\left(\frac{1 + \beta_n}{1 - \alpha_n}\right). \qquad (8)$$

There are two things worth highlighting in Theorem 2. First, if the local models are accurately learned (i.e., $\beta_n = \alpha_n = 0 \,\forall n$), the RHS of Equation 8 equals zero, implying $\pi = \hat{\pi}$. Second, if either $\beta_n \to \infty$ or $\alpha_n \to 1$ for some $n$, the bound in Equation 8 goes to infinity — i.e., it degenerates if one of the local GFlowNets are poorly trained. This is well-aligned with the *catastrophic failure* phenomenon (de Souza et al., 2022), which was originally observed in the literature of parallel MCMC (Neiswanger et al., 2014; Nemeth and Sherlock, 2018; Mesquita et al., 2019) and refers to the incorrectness of the global model due to inadequately estimated local parameters and can result in missing modes or misrepresentation of low-density regions. Figure 1 shows a case where one of the local GFlowNets is poorly trained (Client 2's). Note that minimizing the FB objective leads to a good approximation of the product of marginal distributions over terminal states (encoded by the local GFlowNets). Nonetheless, the result is far from we have envisioned at first, i.e., $R \propto R_1 R_2$.

### 3.2 CONTRASTIVE BALANCE

**Contrastive balance.** As a stepping stone towards proving Theorem 1, we developed the *contrastive balance condition*, a condition that is both necessary and sufficient to guarantee a GFlowNet's marginal over terminal states is proportional to its target reward, as formalized in Lemma 1.

**Lemma 1** (Constrastive balance condition). *If $p_F, p_B \in V^2 \to \mathbb{R}^+$ are the forward and backward policies of a GFlowNet sampling proportionally to some arbitrary reward function $R : \mathcal{X} \to \mathbb{R}^+$ with finite support, then, for any pair of terminal trajectories $\tau, \tau' \in \mathcal{T}$ with $\tau \rightsquigarrow x$ and $\tau \rightsquigarrow x'$,*

$$R(x') \prod_{s \to s' \in \tau} \frac{p_F(s, s')}{p_B(s', s)} = R(x) \prod_{s \to s' \in \tau'} \frac{p_F(s, s')}{p_B(s', s)} \qquad (9)$$

Conversely, if a GFlowNet with forward and backward policies $p_F, p_B$ abide by Equation 9, it induces a marginal distribution over $x \in \mathcal{X}$ proportional to $R$.

*Proof.* Note that CB is satisfied for all $(\tau, \tau')$ iff the quotient $c(\tau) = \frac{p_B(\tau|x)R(x)}{p_F(\tau)}$ does not depend upon $\tau$. This is equivalent to $(p_F, p_B)$ satisfying TB with $\log Z = c(\tau)$ and, hence, the result follows directly from (Malkin et al., 2022, Proposition 1). See Appendix A for a self-contained proof. □

Reviewer: Xj4h

Enforcing Lemma 1 results in a loss that does not depend on estimates of the intractable partition function present in the TB condition. The next corollary guarantees that an instantiation of the GFlowNet parameterized by a global minimizer of $\mathcal{L}_{CB}$ (Eq. 8) correctly samples from $p(x) \propto R(x)$. We call $\mathcal{L}_{CB}$ the contrastive balance loss as it measures the contrast between randomly sampled trajectories. Notably, the expected value of the CB loss is proportional to the variance of a TB-based estimate of the partition function, which has been used as training objective by Zhang et al. (2023a). Additionally, in practice, we observed that in some cases the contrastive balance loss leads to better results than the TB and DB losses, as we will see in Section 4.5.

Reviewer: Xj4h

**Corollary 2** (Contrastive balance loss). *Let $p_F(\cdot, \cdot; \phi_F)$ and $p_B(\cdot, \cdot; \phi_B)$ denote forward/backward policies, and $\nu : \mathcal{T}^2 \to \mathbb{R}^+$ be some full-support probability distribution over pairs of terminal trajectories. Then, $p_T(x; \phi_F) \propto R(x) \, \forall x \in \mathcal{X}$ iff $\mathbb{E}_{(\tau, \tau') \sim \nu}[\mathcal{L}_{CB}(\tau, \tau', \phi_F, \phi_B)] = 0$ where*

$$\mathcal{L}_{CB}(\tau, \tau', \phi_F, \phi_B) = \left( \log \frac{p_F(\tau; \phi_F)}{p_B(\tau; \phi_B)} - \log \frac{p_F(\tau'; \phi_F)}{p_B(\tau'; \phi_B)} + \log \frac{R(x')}{R(x)} \right) \quad (10)$$

To gain further intuition regarding Equation 10, let us define $\xi(\tau; \phi_F, \phi_B)$ as $\xi(\tau; \phi_F, \phi_B) := \log p_F(\tau) - \log R(x) - \log p_B(\tau|x)$. Then, if the GFlowNet is perfectly trained, TB implies $\xi(\tau; \phi_F, \phi_B)$ should be exactly the constant $\log Z = \log \sum_{x \in \mathcal{X}} R(x)$, which does not depend on $\tau$. Furthermore, note that $\mathcal{L}_{CB}(\tau, \tau', \phi_F, \phi_B)$ equals $(\xi(\tau; \phi_F, \phi_B) - \xi(\tau'; \phi_F, \phi_B))^2$. Thus, Equation 9 can be seen as the distance between two "estimates" of $Z$, each based on a terminal trajectory.

**Computational advantages of $\mathcal{L}_{CB}$.** Importantly, note that $\mathcal{L}_{CB}$ incurs learning fewer parameters than TB and DB losses. Besides requiring the forward and backward policies $p_F$ and $p_B$, TB requires parameterizing the partition function of $R$. Alternatively, DB implies using a neural network to approximate the flow through each node. In contrast, CB requires only learning $p_F$ and $p_B$.

$\mathcal{L}_{CB}$ **and VI.** Notably, the next proposition ties the CB loss' gradient to that of a variational objective, extending the characterization of GFlowNets as VI started by Malkin et al. (2023) for the TB loss. More specifically, Theorem 3 states that the on-policy gradients of the CB objective coincide in expectation to the gradient of the KL divergence between the forward and backward policies.

**Theorem 3** (VI and CB). *Let $p_F \otimes p_F$ be the outer product distribution that assings probability $p_F(\tau)p_F(\tau')$ to each pair $(\tau, \tau')$ of trajectories. The criterion in Equation 10 satisfies*

$$\nabla_{\phi_F} \mathcal{D}_{KL}[p_F || p_B] = \frac{1}{4} \mathbb{E}_{(\tau, \tau') \sim p_F \otimes p_F} [\nabla_{\phi_F} \mathcal{L}_{CB}(\tau, \tau', \phi_F, \phi_B)] .$$

*Proof.* Let $\xi_{TB}(\tau; \phi_F, Z) = \log \frac{Z p_F(\tau)}{p_B(\tau|x)R(x)}$. Note $\mathcal{L}_{CB}(\tau, \tau') = (\xi_{TB}(\tau) - \xi_{TB}(\tau'))^2$ and $\xi_{TB}(\tau)^2 = \mathcal{L}_{TB}(\tau)$ is the TB loss. Thus, $\mathbb{E}_{\tau, \tau'}[\nabla_{\phi_F} \mathcal{L}_{CB}(\tau, \tau')] = 2\mathbb{E}_\tau[\nabla_{\phi_F} \xi_{TB}(\tau)^2] - 4\mathbb{E}_\tau[\xi_{TB}(\tau)]\mathbb{E}_\tau[\nabla_{\phi_F} \xi_{TB}(\tau)]$. Also, Prop. 1 of Malkin et al. (2023) ensures $\mathbb{E}_\tau[\nabla_{\phi_F} \xi_{TB}(\tau)] = 0$ and $\mathbb{E}_\tau[\nabla_{\phi_F} \xi_{TB}(\tau)^2] = 2\mathcal{D}_{KL}[p_F || p_B]$, implying our statement. See Appendix A for a self-contained proof. □

Reviewer: Xj4h

## 4 EXPERIMENTS

The main purpose of our experiments is to verify the empirical performance of FC-GFlowNets, i.e., their capacity to accurately sample from the combination of local rewards. To that extent, we consider four diverse tasks: sampling states from a *grid world* in Section 4.1, *generation of multi-sets* (Bengio et al., 2023; Pan et al., 2023a) in Section 4.2, *design of sequences* (Jain et al., 2022) in Section 4.3, and *Bayesian phylogenetic inference* (Zhang and Matsen IV, 2018) in Section 4.4. Since FC-GFlowNet is the first of its kind, we propose two baselines to compare it against: a centralized GFlowNet, which requires clients to openly disclose their rewards, and a divide-and-conquer algorithm in which each client approximates its local GFlowNet with a product of categorical distributions, which is then aggregated via a product in the server. We call the latter approach parallel categorical VI (PCVI). Additionally, Section 4.5 explores the CB criterion as a loss for conventional (non-federated) GFlowNets. Notably, in all experiments, clients contribute with distinct rewards.

Reviewer: wWFS

Table 1: **Quality of the federated approximation** to the combined reward. The table shows i) the L1 distance between the distribution induced by each method and the ground truth and ii) the average log reward of the top-800 scoring samples. Our FC-GFlowNet is consistently better than the PCVI baseline regarding L1 distance, showing approximately the same performance as a centralized GFlowNet. Furthermore, FC-GFlowNet's Top-800 score perfectly matches the centralized model, while PCVI's differ drastically. Values are the average and standard deviation over three repetitions.

| | Grid World | | Multisets | | Sequences | |
| --- | --- | --- | --- | --- | --- | --- |
| | $L_1 \downarrow$ | Top-800 $\uparrow$ | $L_1 \downarrow$ | Top-800 $\uparrow$ | $L_1 \downarrow$ | Top-800 $\uparrow$ |
| Centralized | 0.027 | $-6.355$ | 0.100 | 27.422 | 0.003 | $-1.535$ |
| | $(\pm 0.016)$ | $(\pm 0.000)$ | $(\pm 0.001)$ | $(\pm 0.000)$ | $(\pm 0.001)$ | $(\pm 0.000)$ |
| FC-GFlowNet (**ours**) | **0.038** | **$-6.355$** | **0.130** | **27.422** | **0.005** | **$-1.535$** |
| | $(\pm 0.016)$ | $(\pm 0.000)$ | $(\pm 0.004)$ | $(\pm 0.000)$ | $(\pm 0.002)$ | $(\pm 0.000)$ |
| PCVI | 0.189 | **$-6.355$** | 0.834 | 26.804 | 1.872 | $-16.473$ |
| | $(\pm 0.006)$ | $(\pm 0.000)$ | $(\pm 0.005)$ | $(\pm 0.018)$ | $(\pm 0.011)$ | $(\pm 0.007)$ |

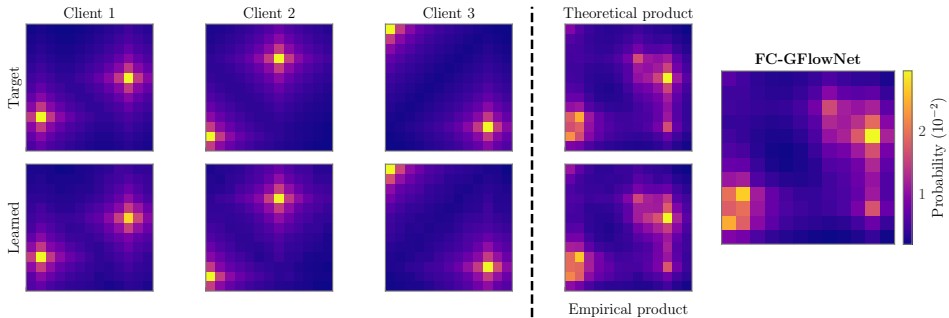

Figure 2: **Grid world.** Each heatmap represents the target distribution (first row), based on the normalized reward, and ones learned by local GFlowNets (second row). Results FC-GFlowNet are in the rightmost pannel. As established by Theorem 1, the good fit of the local models results in an accurate fit to the combined reward.

## 4.1 GRID WORLD

**Task description.** Our grid world environment consists of a Markov decision process over an $18 \times 18$ square grid in which actions consist of choosing a direction ($\rightarrow, \uparrow$) or stopping. The reward for each state $R$ is the sigmoid transform of its minimum distance to a reward beacon (bright yellow in Figure 2). For the federated setting, we consider the problem of combining the rewards from different clients, each of which has two beacons placed in different positions.

**Results.** Figure 2 shows that FC-GFlowNet accurately approximates the targeted distribution, even in cases where combining the client rewards leads to multiple modes. Furthermore, Table 1 shows that FC-GFlowNet performs approximately on par with the centralized model in terms of $L_1$ distance (within one standard deviation), but is three orders of magnitude better than the PCVI baseline. This is also reflected in the average reward over the top 800 samples — identical to the centralized version for FC-GFlowNet, but an order of magnitude smaller for PCVI. Again, these results corroborate our theoretical claims about the correctness of our scheme for combining GFlowNets.

## 4.2 MULTISET GENERATION

**Task description.** Here, the the support of our Reward comprises multisets of size $S$. A multiset $\mathcal{S}$ is built by iteratively including elements from a finite dictionary $U$ to an initially empty set. Each client $n$ assigns a value $r_u^n$ to each word $u \in U$ and defines the log-reward of a multiset $\mathcal{S}$ as the sum of its elements' values; i.e., $\log R_n(\mathcal{S}) = \sum_{u \in \mathcal{S}} r_u^{(n)}$. In practice, the quantities $r_u^{(n)}$ are uniformly picked from the interval $[0, 1]$ for each client. We use $S = 8$ and $|U| = 10$ in our experiments.

**Results.** Figure 3 provides further evidence that our algorithm is able to approximate well the combined reward, even if only given the local GFlowNets. This is further supported by the results in Table 1. Notably, FC-GFlowNet is $\approx 8$ times more accurate than the PCVI baseline.

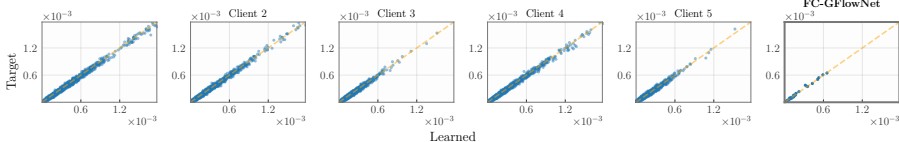

Figure 3: **Multisets: learned × ground truth distributions.** Plots compare normalized rewards vs. distributions learned by GFlowNets. The five plots to the left show local GFlowNets were accurately trained. Thus, a well-trained FC-GFlowNet (right) approximates well the combined reward.

### 4.3 DESIGN OF SEQUENCES

**Task description.** This tasks revolves around building sequences of maximum size $S$. We start with an empty sequence $\mathcal{S}$ and proceed by iteratively appending an element from a fixed dictionary $U$. The process halts when (i) we select a special terminating token or (ii) the sequence length reaches $S$. In the federated setting, we assume each client $n$ has a score $p_s^{(n)}$ to each of the $S$ positions within the sequence and a score $t_u^{(n)}$ to each of the $|U|$ available tokens, yielding the reward of a sequence sequence $\mathcal{S} = (u_1, \ldots, u_M)$ as $R_n(\mathcal{S}) = \exp \sum_{i=1}^{M} p_i^{(n)} t_{u_i}^{(n)}$.

**Results.** Again, Figure 4 follows what we expect given Theorem 1 and shows that FC-GFlowNet accurately samples from the product of individual rewards. Table 1 further reinforces this conclusion, showing a small gap in L1 distance between FC-GFlowNet and the centralized GFlowNet trained with access to all rewards. Notably, our method is $\approx 8$ times more accurate than PCVI. Furthermore, the Top-800 average reward of FC-GFlowNet perfectly matches the centralized model.

Figure 4: **Sequences: learned × ground truth distributions.** . Plots compare normalized rewards to distributions learned by GFlowNets. The five leftmost plots show local GFlowNets were well trained. Hence, as implied by Theorem 1, FC-GFlowNet approximates well the combined reward.

### 4.4 BAYESIAN PHYLOGENETIC INFERENCE

**Task description.** In this task, we are interested on inferring a *phylogeny* $T = (t, b)$, which is a characterization of the evolutionary relationships between biological species and is composed by a tree topology $t$ and its $(2N - 1)$-dimensional vector of non-negative branch lengths $b$. The topology $t$ is as a leaf-labeled complete binary tree with $N$ leaves, each corresponding to a species. Notably, $T$ induces a probability distribution $P$ over the space of nucleotide sequences $Y_1, \ldots, Y_M \in \Omega^N$, where $\Omega$ is a vocabulary of *nucleobases* and $Y_m$ denote the nucleobases observed at the $m$-th site for each species. Assume $t$ is a rooted in some node $r$ and that $\pi \in \Delta^{|\Omega|}$ is the prior probability distribution over the nucleobases' frequencies at $r$. Then, the marginal likelihood of a nucleobase $Y_m$ occurring *site m* for node $n$ is recursively defined by (Felsenstein, 1981) as

$$\mathbf{P}_n(Y_m|T) = \begin{cases} \text{One-Hot}(Y_{m,n}) & \text{if } n \text{ is a leaf,} \\ \left[\left(e^{b_{n,n_l}Q}\mathbf{P}_{n_l}(Y_m|T)^\intercal\right) \odot \left(e^{b_{n,n_r}Q}\mathbf{P}_{n_r}(Y_m|T)^\intercal\right)\right]^\intercal & \text{otherwise,} \end{cases}$$

in which $n_l$ and $n_r$ are respectively the left and right children of $n$; $b_{n,a}$ is the length of the branch between nodes $n$ and $a$; and $Q \in \mathbb{R}^{|\Omega| \times |\Omega|}$ is an instantaneous rate-conversion matrix for the underlying substitution rates between nucleotides, which is given beforehand. In this context, the marginal likelihood of the observed data within the site m is $\mathbf{P}_r(Y_m|T)^\intercal \pi$ and, assuming conditional independence of the sites given $T$, the overall likelihood of the data is $\mathbf{P}(\mathbf{Y}|T) = \prod_{1 \le i \le M}(\mathbf{P}(Y_i|T)^T \pi)$ — which is naturally log-additive on the sites. For our experiments, $\pi$ is a uniform distribution. For simplicity, we consider constant branch length, fixed throughout the experiments. For the federated setting, we place a uniform prior over $t$ and split 2500 nucleotide sites across five clients. In parallel MCMC (Neiswanger et al., 2014) fashion, each client trains a GFlowNet to sample from its local posterior, proportional to the product of its local likelihood and a scaled version of the prior.

**Results.** Figure 5 shows that Federated GFlowNet accurately learns the posterior distribution over the tree's topologies: the $L_1$ error between the learned distribution and the targeted product distribution is $0.088$, whereas the average $L_1$ error among the clients is $0.083(\pm 0.041)$. Noticeably, this indicates the model's aptitude to learn a highly sparse posterior distribution in a decentralized manner. Moreover, our results suggest the potential usefulness of GFlowNets as an alternative to the notoriously inefficient MCMC-based algorithms (Zhang and Matsen IV, 2018) in the field of evolutionary biology. Importantly, our method is also the first provably correct algorithm enabling distributed Bayesian inference over discrete objects, which may be invaluable in real-world problems with millions of sites. Notably, naive strategies, like the PCVI baseline, consistently lead to sampling elements that do not belong to the support of our posterior (i.e., are invalid) — which is why we do not compare against it. In future endeavors, we plan to investigate joint parallel inference on the tree's topology and its branch' lengths using hybrid-space GFlowNets (Deleu et al., 2023).

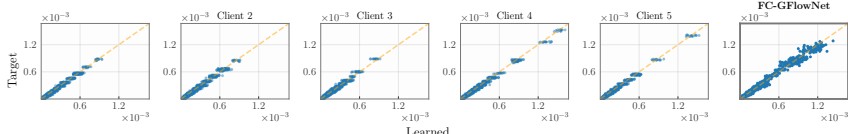

Figure 5: **Bayesian phylogenetic inference: learned $\times$ ground truth distributions.** Following the pattern in Figures 2-4, the goodness-of-fit from local GFlowNets (Clients 1-5) is directly reflected in the distribution learned by FC-GFlowNet.

### 4.5 EVALUATING THE CB LOSS

Section 3.2 presents the CB loss as a natural development given the theory of FC-GFlowNets. To evaluate its utility as a criterion to train GFlowNets in the conventional centralized (non-federated) setting, we report the evolution during training of the L1 error of the GFlowNet wrt the normalized reward for models trained using DB, TB, and our CB. We do so for all tasks in our experiments, with all GFlowNets using the same architectures for the forward and backward policies (more details in supplement). While we did not see any noticeable difference for the tasks in which all states are also terminal (grid world and design of sequences), CB led to better convergence in

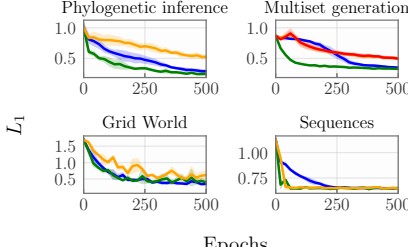

Figure 6: $\mathcal{L}_{CB}$ performs competitively or better than $\mathcal{L}_{TB}$, $\mathcal{L}_{DB}$ and $\mathcal{L}_{FL}$.

the multiset generation and phylogeny tasks (Figure 6). An explanation is that CB incurs a considerably simpler parameterization than DB and TB — as we do not require estimating the flow nor the partition function. In practice, it can be challenging to optimize the log partition in TB. Therefore, our appendix shows more results comparing the CB to TB w/ different learning rates for $\log Z_{\phi_Z}$ in the multiset experiments (Figure 8). Noticeably, CB outperforms TB for all rates we have tested. A rigorous understanding, however, of the diversely proposed balance conditions is still lacking in the literature and is an important course of action for future research.

**Reviewer:** Xj4h

## 5 CONCLUSIONS

We proposed FC-GFlowNet as a simple and elegant solution for federated learning of GFlowNets, which we validate on a suite of experiments. Our method enjoys theoretical guarantees and builds on the concept of contrastive balance (CB). Our theoretical analysis i) guarantees correctness when local models are perfectly trained and ii) allows us to quantify the impact of errors of local models on the federated one. Additionally, we observed that using CB loss led to faster convergence for the local clients when intermediate states are not terminal — while being competitive in other scenarios.

We believe FC-GFlowNets pave the way for a range of applications of federated Bayesian inference over discrete parameter spaces. We also believe FC-GFlowNets might be useful to scale up Bayesian inference by amortizing the cost of expensive likelihood computations over different clients. In the realm of multi-objective optimization, FC-GFlowNets enable sampling from a combination of rewards by leveraging pre-trained GFlowNets — even without direct access to the rewards, as recently explored by Garipov et al. (2023).

**Reviewer:** Xj4h

## REPRODUCIBILITY STATEMENT

The experiments in this work are reproducible. Upon acceptance, we will make our code publicly available on GitHub.

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

## A PROOFS

### A.1 PROOF OF LEMMA 1

It stems directly from the trajectory balance that, for any trajectory $\tau^\star \in \mathcal{T}$:

$$Z \prod_{s \to s' \in \tau^\star} p_F(s \to s') = R(x) \prod_{s \to s' \in \tau^\star} p_B(s' \to s) \tag{11}$$

$$\iff Z = R(x) \prod_{s \to s' \in \tau^\star} \frac{p_B(s' \to s)}{p_F(s \to s')} \tag{12}$$

Therefore, applying this identity to $\tau$ and $\tau'$ and equating the right-hand-sides (RHSs) yields Equation 9. We are left with the task of proving the converse. Note we can rewrite Equation 9 as:

$$R(x) \prod_{s \to s' \in \tau} \frac{p_B(s' \to s)}{p_F(s \to s')} = R(x') \prod_{s \to s' \in \tau'} \frac{p_B(s' \to s)}{p_F(s \to s')}. \tag{13}$$

If Equation 9 holds for any pair $(\tau, \tau')$, we can vary $\tau'$ freely for a fixed $\tau$ — which implies the RHS of the above equation must be a constant with respect to $\tau'$. Say this constant is $c$, then:

$$R(x) \prod_{s \to s' \in \tau} \frac{p_B(s' \to s)}{p_F(s \to s')} = c \tag{14}$$

$$\iff R(x) \prod_{s \to s' \in \tau} p_B(s' \to s) = c \prod_{s \to s' \in \tau} p_F(s \to s'), \tag{15}$$

and summing the above equation over all $\tau \in \mathcal{T}$ yields:

$$\sum_{\tau \in \mathcal{T}} R(x) \prod_{s \to s' \in \tau} p_B(s' \to s) = c \sum_{\tau \in \mathcal{T}} \prod_{s \to s' \in \tau} p_F(s \to s') \tag{16}$$

$$\implies \sum_{\tau \in \mathcal{T}} R(x) \prod_{s \to s' \in \tau} p_B(s' \to s) = c \tag{17}$$

Furthermore, note that:

$$\sum_{x \in \mathcal{X}} R(x) \sum_{\tau \in T(x)} \prod_{s \to s' \in \tau} p_B(s' \to s) = c \tag{18}$$

$$\implies \sum_{x \in \mathcal{X}} R(x) = c \tag{19}$$

$$\implies Z = c \tag{20}$$

Plugging $Z = c$ into Equation 14 yields the trajectory balance condition.

### A.2 PROOF OF THEOREM 1

The proof is based on the following reasoning. We first show that, given the satisfiability of the federated balance condition, the marginal distribution over the terminating states is proportional to

$$\mathbb{E}_{\tau \sim p_B(\cdot|x)} \left[ \prod_{1 \le i \le N} \frac{p_F^{(i)}(\tau)}{p_B^{(i)}(\tau|x)} \right], \tag{21}$$

as stated in Remark 1. Then, we verify that this distribution is the same as

$$p_T(x) \propto \prod_{1 \le i \le N} R_i(x) \tag{22}$$

if the local balance conditions are satisfied. This proves the sufficiency of the federated balance condition for building a model that samples from the correct product distribution. The necessity follows from Proposition 16 of Bengio et al. (2023) and from the observation that the local balance conditions are equivalent to $p_F^{(i)}(\tau)/p_B^{(i)}(\tau|x) = R_i(x)$ for each $i = 1, \ldots, N$.

Next, we provide a more detailed discussion about this proof. Similarly to subsection A.1, notice that the contrastive nature of the federated balance condition implies that, if

$$\prod_{1 \leq i \leq N} \frac{\left( \prod_{s \to s' \in \tau} \frac{p_F^{(i)}(s,s')}{p_B^{(i)}(s',s)} \right)}{\left( \prod_{s \to s' \in \tau'} \frac{p_F^{(i)}(s,s')}{p_B^{(i)}(s',s)} \right)} = \frac{\left( \prod_{s \to s' \in \tau} \frac{p_F(s,s')}{p_B(s',s)} \right)}{\left( \prod_{s \to s' \in \tau'} \frac{p_F(s,s')}{p_B(s',s)} \right)}, \tag{23}$$

then

$$p_F(\tau) = c \left( \prod_{1 \leq i \leq N} \frac{p_F^{(i)}(\tau)}{p_B^{(i)}(\tau|x)} \right) p_B(\tau|x) \tag{24}$$

for a constant $c > 0$ that does not depend either on $x$ or on $\tau$. Hence, the marginal distribution over a terminating state $x \in \mathcal{X}$ is

$$p_T(x) := \sum_{\tau \leadsto x} \prod_{s \to s' \in \tau} p_F(s \to s') \tag{25}$$

$$= c \sum_{\tau \leadsto x} \left( \prod_{1 \leq i \leq N} \frac{p_F^{(i)}(\tau)}{p_B^{(i)}(\tau|x)} \right) p_B(\tau|x) \tag{26}$$

$$= c \mathbb{E}_{\tau \sim p_B(\cdot|x)} \left[ \prod_{1 \leq i \leq N} \frac{p_F^{(i)}(\tau)}{p_B^{(i)}(\tau|x)} \right]. \tag{27}$$

Correspondingly, $p_F^{(i)}(\tau)/p_B^{(i)}(\tau|x) \propto R_i(x)$ for every $i = 1, \ldots, N$ and every $\tau$ leading to $x$ due to the satisfiability of the local balance conditions. Thus,

$$p_T(x) \propto \mathbb{E}_{\tau \sim p_B(\cdot|x)} \left[ \prod_{1 \leq i \leq N} R_i(x) \right] = \prod_{1 \leq i \leq N} R_i(x), \tag{28}$$

which attests the sufficiency of the federated balance condition for the distributional correctness of the global model.

## A.3 PROOF OF THEOREM 2

Initially, recall that the Jeffrey divergence, known as the symmetrized KL divergence, is defined as

$$\mathcal{D}_J(p, q) = \mathcal{D}_{KL}[p||q] + \mathcal{D}_{KL}[q||p] \tag{29}$$

for any pair $p$ and $q$ of equally supported distributions. Then, let

$$\hat{\pi}(x) = \hat{Z} \, \mathbb{E}_{\tau \sim p_B(\cdot|x)} \left[ \prod_{1 \leq i \leq N} \frac{p_F^{(i)}(\tau)}{p_B^{(i)}(\tau|x)} \right] \tag{30}$$

be the marginal distribution over the terminating states of a GFlowNet satisfying the federated balance condition (see Remark 1 and subsection A.2). On the one hand, notice that

$$\mathcal{D}_{KL}[\pi||\hat{\pi}] = \mathbb{E}_{x\sim\pi}\left[\log\frac{\pi(x)}{\hat{\pi}(x)}\right] \tag{31}$$

$$= \mathbb{E}_{x\sim\pi}\left[\log\pi(x) - \log Z\mathbb{E}_{\tau\sim p_B(\cdot|x)}\left[\prod_{1\leq i\leq N}\frac{p_F^{(i)}(\tau)}{p_B^{(i)}(\tau|x)}\right]\right] \tag{32}$$

$$= -\mathbb{E}_{x\sim\pi}\left[\log\mathbb{E}_{\tau\sim p_B(\cdot|x)}\left[\prod_{1\leq i\leq N}\frac{p_F^{(i)}(\tau)}{p_B^{(i)}(\tau|x)\pi_i(x)}\right]\right] - \log\hat{Z} + \log Z \tag{33}$$

$$\leq -\mathbb{E}_{x\sim\pi}\left[\log\prod_{1\leq i\leq N}(1-\alpha_i)\right] - \log\hat{Z} + \log Z \tag{34}$$

$$= \log\frac{Z}{\hat{Z}} + \sum_{1\leq i\leq N}\log\left(\frac{1}{1-\alpha_i}\right), \tag{35}$$

in which $Z := \left(\sum_{x\in\mathcal{X}}\prod_{1\leq i\leq N}\pi_i(x)\right)^{-1}$ is $\pi$'s normalization constant. On the other hand,

$$\mathcal{D}_{KL}[\pi||\hat{\pi}] = \mathbb{E}_{x\sim\hat{\pi}}\left[\log\frac{\hat{\pi}(x)}{\pi(x)}\right] \tag{36}$$

$$= \mathbb{E}_{x\sim\hat{\pi}}\left[\log Z\mathbb{E}_{\tau\sim p_B(\cdot|x)}\left[\prod_{1\leq i\leq N}\frac{p_F^{(i)}(\tau)}{p_B^{(i)}(\tau|x)}\right] - \log\pi(x)\right] \tag{37}$$

$$= \mathbb{E}_{x\sim\hat{\pi}}\left[\log\mathbb{E}_{\tau\sim p_B(\cdot|x)}\left[\prod_{1\leq i\leq N}\frac{p_F^{(i)}(\tau)}{p_B^{(i)}(\tau|x)\pi_i(x)}\right]\right] + \log\hat{Z} - \log Z \tag{38}$$

$$\leq \mathbb{E}_{x\sim\hat{\pi}}\left[\log\prod_{1\leq i\leq N}(1+\beta_i)\right] + \log\hat{Z} - \log Z \tag{39}$$

$$= \log\frac{\hat{Z}}{Z} + \sum_{1\leq i\leq N}\log(1+\beta_i). \tag{40}$$

Thus, the Jeffrey divergence between the targeted product distribution $\pi$ and the effectively learned distribution $\hat{\pi}$ is

$$\mathcal{D}_J(\pi,\hat{\pi}) = \mathcal{D}_{KL}[\pi||\hat{\pi}] + \mathcal{D}_{KL}[\hat{\pi}||\pi] \tag{41}$$

$$\leq \log\frac{Z}{\hat{Z}} + \sum_{1\leq i\leq N}\log\left(\frac{1}{1-\alpha_i}\right) + \log\frac{\hat{Z}}{Z} + \sum_{1\leq i\leq N}\log(1+\beta_i) \tag{42}$$

$$= \sum_{1\leq i\leq N}\log\left(\frac{1+\beta_i}{1-\alpha_i}\right). \tag{43}$$

### A.4 PROOF OF THEOREM 3

We firstly recall the construction of the unbiased REINFORCE gradient estimator (Williams 1992), which was originally designed as a method to implement gradient-ascent algorithms to tackle associative tasks involving stochastic rewards in reinforcement learning. Let $p_\theta$ be a probability density (or mass function) differentiably parametrized by $\theta$ and $f_\theta\colon\mathcal{X}\to\mathbb{R}$ be a real-value function over $\mathcal{X}$ possibly dependent on $\theta$. Our goal is to estimate the gradient

$$\nabla_\theta\mathbb{E}_{x\sim p_\theta}[f_\theta(x)], \tag{44}$$

which is not readily computable due to the dependence of $p_\theta$ on $\theta$. However, since

$$\nabla_\theta \mathbb{E}_{x \sim p_\theta}[f_\theta(x)] = \nabla_\theta \int_{x \in \mathcal{X}} f_\theta(x) p_\theta(x) \mathrm{d}x \tag{45}$$

$$= \int_{x \in \mathcal{X}} ((\nabla_\theta f_\theta(x)) p_\theta(x)) \, \mathrm{d}x + \int_{x \in \mathcal{X}} ((\nabla_\theta p_\theta(x)) f_\theta(x)) \, \mathrm{d}x \tag{46}$$

$$= \mathbb{E}_{x \sim p_\theta} \left[ \nabla_\theta f_\theta(x) + f_\theta(x) \nabla_\theta \log p_\theta(x) \right], \tag{47}$$

the gradient of $f_\theta$'s expected value under $p_\theta$ may be unbiasedly estimated by averaging the quantity $\nabla_\theta f_\theta(x) + f_\theta(x) \nabla_\theta \log p_\theta(x)$ over samples of $p_\theta$. We use this identity to compute the KL divergence between the forward and backward policies of a GFlowNet. In this sense, notice that

$$\nabla_\theta \mathcal{D}_{KL}[p_F || p_B] = \nabla_\theta \mathbb{E}_{\tau \sim p_F} \left[ \log \frac{p_F(\tau)}{p_B(\tau)} \right] \tag{48}$$

$$= \mathbb{E}_{\tau \sim p_F} \left[ \nabla_\theta \log p_F(\tau) + \left( \log \frac{p_F(\tau)}{p_B(\tau)} \right) \nabla_\theta \log p_F(\tau) \right] \tag{49}$$

$$= \mathbb{E}_{\tau \sim p_F} \left[ \left( \log \frac{p_F(\tau)}{p_B(\tau)} \right) \nabla_\theta \log p_F(\tau) \right], \tag{50}$$

as $\mathbb{E}_{\tau \sim p_F}[\nabla_\theta \log p_F(\tau)] = \nabla_\theta \mathbb{E}_{\tau \sim p_F}[1] = 0$. In contrast, the gradient of the contrastive balance loss with respect to $\theta$ is

$$\nabla_\theta \mathcal{L}_{CB}(\tau, \tau', \theta) = \nabla_\theta \left( \log \frac{p_F(\tau)}{p_B(\tau)} - \log \frac{p_F(\tau')}{p_B(\tau')} \right)^2 \tag{51}$$

$$= 2 \left( \log \frac{p_F(\tau)}{p_B(\tau)} - \log \frac{p_F(\tau')}{p_B(\tau')} \right) (\nabla_\theta \log p_F(\tau) - \nabla_\theta \log p_F(\tau')), \tag{52}$$

whose expectation under the outer product distribution $p_F \otimes p_F$ equals the quantity $4 \nabla_\theta \mathcal{D}_{KL}[p_F || p_B]$ in Equation 48. Indeed, as

$$\mathbb{E}_{\tau \sim p_F} \left[ \left( \log \frac{p_F(\tau')}{p_B(\tau')} \right) \nabla_\theta \log p_F(\tau) \right] = 0, \tag{53}$$

with an equivalent identity obtained by interchanging $\tau$ and $\tau'$,

$$\mathbb{E}_{(\tau, \tau') \sim p_F \otimes p_F} \left[ \nabla_\theta \mathcal{L}_{CB}(\tau, \tau', \theta) \right] = \tag{54}$$

$$\mathbb{E}_{(\tau, \tau') \sim p_F \otimes p_F} \left[ 2 \left( \log \frac{p_F(\tau)}{p_B(\tau)} - \log \frac{p_F(\tau')}{p_B(\tau')} \right) (\nabla_\theta \log p_F(\tau) - \nabla_\theta \log p_F(\tau')) \right] = \tag{55}$$

$$\mathbb{E}_{(\tau, \tau') \sim p_F \otimes p_F} \left[ 2 \left( \log \frac{p_F(\tau)}{p_B(\tau)} \right) \nabla_\theta \log p_F(\tau) + 2 \left( \log \frac{p_F(\tau')}{p_B(\tau')} \right) \nabla_\theta \log p_F(\tau') \right] = \tag{56}$$

$$\mathbb{E}_{\tau \sim p_F} \left[ 4 \left( \log \frac{p_F(\tau)}{p_B(\tau)} \right) \nabla_\theta \log p_F(\tau) \right] = 4 \nabla_\theta \mathcal{D}_{KL}[p_F || p_B]. \tag{57}$$

Thus, the on-policy gradient of the contrastive balance loss equals in expectation the gradient of the KL divergence between the forward and backward policies of a GFlowNet.

## B    EXPONENTIALLY WEIGHTED DISTRIBUTIONS

This section extends our theoretical results and shows how to train a FC-GFlowNet to sample from a logarithmic pool of locally trained GFlowNets. Henceforth, let $R_1, \ldots, R_N \colon \mathcal{X} \to \mathbb{R}_+$ be non-negative functions over $\mathcal{X}$ and assume that each client $n = 1, \ldots, N$ trains a GFlowNet to sample proportionally to $R_n$. The next propositions show how to train a GFlowNet to sample proportionally to an exponentially weighted distribution $\prod_{n=1}^N R_n(x)^{\omega_n}$ for non-negative weights $\omega_1, \ldots, \omega_N$. We omit the proofs since they are essentially identical to the ones presented in Appendix A.

Firstly, Theorem 1$'$ below proposes a modified balance condition for the global GFlowNet and shows that the satisfiability of this condition leads to a generative model that samples proportionally to the exponentially weighted distribution.

---

**Algorithm 1** Training of Federated GFlowNets

---

**Require:** $\left(p_F^{(1)}, p_B^{(1)}\right), \ldots, \left(p_F^{(K)}, p_B^{(K)}\right)$ clients' policies, $R_1, \ldots, R_K$ clients' rewards, $(p_F, p_B)$ parameterized global policies, $E$ number of epochs for training, $u_F$ uniform policy

**Ensure:** $p^{\intercal}(x) \propto R(x) := \prod_{1 \leq k \leq K} R_k(x)$

    **parfor** $k \in \{1, \ldots, K\}$ **do**                     ▷ Train the clients' models in parallel

        train the policies $\left(p_F^{(k)}, p_B^{(k)}\right)$ to sample proportionally to $R_k$

    **end parfor**

    **for** $e \in \{1, \ldots, E\}$ **do**                            ▷ Train the global model

        $\mathcal{B} \leftarrow \{(\tau, \tau'): \tau, \tau' \sim \nicefrac{1}{2} \cdot p_F + \nicefrac{1}{2} \cdot u_F\}$        ▷ Sample a batch of trajectories

        $L \leftarrow \frac{1}{|\mathcal{B}|} \sum_{\tau, \tau' \in \mathcal{B}} \mathcal{L}_{FB}\left(\tau, \tau'; \left\{\left(p_F^{(1)}, p_B^{(1)}\right), \ldots, \left(p_F^{(K)}, p_B^{(K)}\right)\right\}\right)$

        Update the parameters of $p_F$ and $p_B$ through gradient descent on $L$

    **end for**

---

**Theorem 1′** (Federated balance condition). *Let $\left(p_F^{(1)}, p_F^{(1)}\right), \ldots, \left(p_F^{(N)}, p_F^{(N)}\right) : V^2 \to \mathbb{R}^+$ be pairs of forward and backward policies from $N$ GFlowNets sampling respectively proportional to $R_1, \ldots, R_N : \mathcal{X} \to \mathbb{R}^+$. Then, another GFlowNet with forward and backward policies $p_F, p_B \in V^2 \to \mathbb{R}^+$ samples proportionally to $R(x) := \prod_{n=1}^{N} R(x)^{\omega_n}$ if and only if the following condition holds for any terminal trajectories $\tau, \tau' \in \mathcal{T}$:*

$$\prod_{1 \leq i \leq N} \frac{\left(\prod_{s \to s' \in \tau} \frac{p_F^{(i)}(s,s')}{p_B^{(i)}(s',s)}\right)^{\omega_i}}{\left(\prod_{s \to s' \in \tau'} \frac{p_F^{(i)}(s,s')}{p_B^{(i)}(s',s)}\right)^{\omega_i}} = \frac{\left(\prod_{s \to s' \in \tau} \frac{p_F(s,s')}{p_B(s',s)}\right)}{\left(\prod_{s \to s' \in \tau'} \frac{p_F(s,s')}{p_B(s',s)}\right)}. \tag{58}$$

Secondly, Theorem 2′ provides an upper bound on the discrepancy between the targeted and the learned global distributions under controlled local errors — when the local distributions are heterogeneously pooled. Notably, it suggests that the effect of the local failures over the global approximation may be mitigated by reducing the weights associated with improperly trained local models.

**Theorem 2′** (Influence of local failures). *Let $\pi_n := R_n/Z_n$ and $p_F^{(n)}$ and $p_B^{(n)}$ be the forward and backward policies of the $n$th client. We use $\tau \leadsto x$ to indicate that $\tau \in \mathcal{T}$ is finished by $x \to s_f$. Suppose that the local balance conditions are lower- and upper-bounded $\forall n = 1, \ldots, N$ as per*

$$1 - \alpha_n \leq \min_{x \in \mathcal{X}, \tau \leadsto x} \frac{p_F^{(n)}(\tau)}{p_B^{(n)}(\tau|x)\pi_n(x)} \leq \max_{x \in \mathcal{X}, \tau \leadsto x} \frac{p_F^{(n)}(\tau)}{p_B^{(n)}(\tau|x)\pi_n(x)} \leq 1 + \beta_n \tag{59}$$

*where $\alpha_n \in (0,1)$ and $\beta_n > 0$. The Jeffrey divergence $\mathcal{D}_J$ between the global model $\hat{\pi}(x)$ that fulfills the federated balance condition in Equation 4 and $\pi(x) \propto \prod_{n=1}^{N} \pi_n(x)^{\omega_n}$ then satisfies*

$$\mathcal{D}_J(\pi, \hat{\pi}) \leq \sum_{n=1}^{N} \omega_n \log\left(\frac{1 + \beta_n}{1 - \alpha_n}\right). \tag{60}$$

Interestingly, one could train a *conditional* GFlowNet (Bengio et al., 2021) to build an amortized generative model able to sample proportionally to $\prod_{n=1}^{N} R_n(x)^{\omega_n}$ for any non-negative weights $(\omega_1, \ldots, \omega_N)$ within a prescribed set. This is a promising venue for future research.

## C   ADDITIONAL EXPERIMENTS AND IMPLEMENTATION DETAILS

This section is organized as follows. First, Appendix C.1 describes the experimental setup underlying the empirical evaluation of FC-GFlowNets in section 4. Second, Appendix C.2 exhibits the details of the variational approximations to the combined distributions used as baselines in Table 1. Third, Appendix C.3 specifies our settings for comparing the training convergence speed of different optimization objectives. Algorithm 1 illustrates the training procedure of Federated GFlowNets.

**Reviewer:** wWFS

Figure 7: **An illustration of the generative process for phylogenetic trees' topologies.** We iteratively select two trees to join their roots. The final state corresponds to a single, connected graph.

## C.1   EXPERIMENTAL SETUP

In the following, we applied the same optimization settings for each environment. For the stochastic optimization, we minimized the contrastive balance objective using the AdamW optimizer (Loshchilov and Hutter, 2019) for both local and global GFlowNets. We trained the models for 5000 epochs (20000 for the grid world) with a learning rate equal to $3 \cdot 10^{-3}$ with a batch size dependent upon the environment. Correspondingly, we define the $L_1$ error between the distributions $\pi$ and $\hat{\pi}$ as two times the total variation distance between them, $\|\pi - \hat{\pi}\|_1 := \sum_{x \in \mathcal{X}} |\pi(x) - \hat{\pi}(x)|$. For the grid world and design of sequences setups, all intermediate GFlowNet states are also terminal, since they lie on the path from the initial state to another terminal state. For the remaining setups, the intersection between terminal and intermediate states is empty.

`Reviewer: rsSg`

**Grid world.** We considered a two-dimensional grid with length size 12 as the environment for the results of both Table 1 and Figure 2. To parametrize the forward policy, we used an MLP with two 64-dimensional layers and a LeakyReLU activation function between them (Maas et al., 2013). For inference, we simulated $10^6$ environments to (i) compute the $L_1$ error between the targeted and the learned distributions. and (ii) select the 800 most rewarding samples. We utilized a batch size equal to 1024 during both the training and inference phases.

**Design of sequences.** We trained the GFlowNets to generate sequences of size up to 6 with elements selected from a set of size 6. We parametrized the forward policies with a single 64-dimensional layer bidirectional LSTM network followed by an MLP with two 64-dimensional layers (Graves and Graves, 2012). For training, we used a batch size of 512. For inference, we increased the batch size to 1024 and we sampled $10^6$ sequences to estimate the quantities reported in Table 1 and Figure 4.

**Multiset generation.** We designed the GFlowNet to generate multisets of size 8 by iteratively selecting elements from a set $U$ of size 10. Moreover, we endowed each element within $U$ with a learnable and randomly initialized 10-dimensional embedding. To estimate the transition probabilities at a given state $s$, we applied an MLP with two 64-dimensional layers to the sum of the embeddings of the elements in $s$. During training, we used a batch size of 512 to parally generate multiple multisets and reduce the noiseness of the backpropagated gradients. During inference, we increased the batch size to 1024 and generated $10^6$ samples to generate the results reported in Table 1 and Figure 3.

**Bayesian phylogenetic inference.** We devised a GFlowNet to learn a posterior distribution over the space of rooted phylogenetic trees with 7 leaves and fixed branch lengths. Each state is represented as a forest. Initially, each leaf belongs to a different singleton tree. An action consists of picking two trees and joining their roots to a newly added node. The generative process is finished when all nodes are connected in a single tree (see Figure 7). To estimate the policies at the (possibly partially built) tree $t$, we used a graph isomorphism network (GIN; Xu et al., 2019) with two 64-dimensional layers to generate node-level representations for $t$ and then used an MLP to project the sum of these representations to a probability distribution over the viable transitions at $t$. We used a tempered version of the likelihood to increase the sparsity of the targeted posterior. Importantly, we selected a batch size of 512 for training and of 1024 for inference. Results for Table 1 and Figure 5 are estimates based on $10^5$ trees drawn from the learned distributions.

`Reviewer: Xj4h`

Our implementations were based on `PyTorch` (Paszke et al., 2019) and on `PyTorch Geometric` (Fey and Lenssen, 2019).

### C.2 Parallel Categorical Variational Inference

As a simplistic approach to combining the locally learned distributions over compositional objects, we variationally approximate them as the product of categorical distributions over the objects' components. For this, we select the parameters that minimize the reverse Kullback-Leibler divergence between the GFlowNet's distribution $p_T$ and the variational family $\mathcal{Q}$,

$$\hat{q} = \arg\min_{q \in \mathcal{Q}} \mathrm{KL}[p_T || q] = \arg\min_{q \in \mathcal{Q}} -\mathbb{E}_{x \sim p_T}[\log q(x)], \tag{61}$$

which, in asymptotic terms, is equivalent to choosing the parameters that maximize the likelihood of the GFlowNet's samples under the variational model. Then, we use a logarithmic pool of these local variational approximations as a proxy for the global model. In the next paragraphs, we present the specific instantiations of this method for the domains we considered throughout our experiments. We used the same experimental setup of subsection C.1 to train the local GFlowNets.

**Grid world.** An object in this domain is composed of its two coordinates in the grid. For a grid of width $W$ and height $H$, we consider the variational family

$$\mathcal{Q} = \{(\phi, \psi) \in \Delta^{W+1} \times \Delta^{H+1} \colon q_{\phi,\psi}(x, y) = \mathrm{Cat}(x|\phi)\mathrm{Cat}(y|\psi)\}, \tag{62}$$

in which $\Delta^d$ is the $d$-dimensional simplex and $\mathrm{Cat}(\phi)$ ($\mathrm{Cat}(\psi)$) is a categorical distribution over $\{0, \ldots, W\}$ ($\{0, \ldots, H\}$) parametrized by $\phi$ ($\psi$). Then, given the $N$ variational approximations $\left(q_{\phi^{(1)},\psi^{(1)}}\right), \ldots, \left(q_{\phi^{(N)},\psi^{(N)}}\right)$ individually adjusted to the distributions learned by the local GFlowNets, we estimate the unnormalized parameters $\tilde{\phi}$ and $\tilde{\psi}$ of the variational approximation to the global distribution over the positions within the grid as

$$\tilde{\phi} = \bigodot_{1 \leq i \leq N} \phi^{(i)} \text{ and } \tilde{\psi} = \bigodot_{1 \leq i \leq N} \psi^{(i)}. \tag{63}$$

Then, we let $\phi = \phi_u/\phi_u^\intercal \mathbf{1}_{W+1}$ and $\psi = \psi_u/\psi_u^\intercal \mathbf{1}_{H+1}$, with $\mathbf{1}_d$ as the d-dimensional vector of 1s, be the parameters of the global model.

**Design of sequences.** We represent sequences of size up to $T$ over a dictionary $V$ as a tuple $(S, (x_1, \ldots, x_S))$ denoting its size $S$ and the particular arrangement of its elements $(x_1, \ldots, x_S)$. This is inherently modeled as a hierarchical model of categorical distributions,

$$S \sim \mathrm{Cat}(\theta), \tag{64}$$
$$x_i \sim \mathrm{Cat}(\phi_{i,S}|S) \text{ for } i \in \{1, \ldots, S\}, \tag{65}$$

which is parametrized by $\theta \in \Delta^T$ and $\phi_{\cdot,S} \in \mathbb{R}^{S \times |V|}$ for $S \in \{1, \ldots, T\}$. We define our family of variational approximations as the collection of all such hierarchical models and estimate the parameters $\theta$ and $\phi$ accordingly to Equation 61. In this case, let $\left(\theta^{(1)}, \phi^{(1)}\right), \ldots, \left(\theta^{(N)}, \phi^{(N)}\right)$ be the parameters associated with the variational approximations to each of the $N$ locally trained GFlowNets. The unnormalized parameters $\tilde{\theta}$ and $\tilde{\phi}$ of the combined model that approximates the global distribution over the space of sequences are then

$$\tilde{\theta} = \bigodot_{1 \leq i \leq N} \theta^{(i)} \text{ and } \tilde{\phi}_{\cdot,S} = \bigodot_{1 \leq i \leq N} \phi^{(i)}_{\cdot,S} \text{ for } S \in \{1, \ldots, T\}, \tag{66}$$

whereas the normalized ones are $\theta = \tilde{\theta}/\tilde{\theta}^\intercal \mathbf{1}_T$ and $\phi_{\cdot,S} = \mathrm{diag}(\tilde{\phi}_{\cdot,S}\mathbf{1}_{|V|})^{-1}\tilde{\phi}_{\cdot,S}$.

**Multiset generation.** We model a multiset $\mathcal{S}$ of size $S$ as a collection of independently sampled elements from a warehouse $\mathcal{W}$ with replacement. This characterizes the variational family

$$\mathcal{Q} = \left\{ q(\cdot|\phi) \colon q(\mathcal{S}|\phi) = \prod_{s \in \mathcal{S}} \mathrm{Cat}(s|\phi) \right\}, \tag{67}$$

in which $\phi$ is the parameter of the categorical distribution over $\mathcal{W}$ estimated through Equation 61. Denote by $\phi^{(1)}, \ldots, \phi^{(N)}$ the estimated parameters that disjointly approximate the distribution of $N$ locally trained GFlowNets. We then variationally approximate the logarithmically pooled global distribution as $q(\cdot|\phi) \in \mathcal{Q}$ with $\phi = \tilde{\phi}/\tilde{\phi}^\intercal \mathbf{1}_{|\mathcal{W}|}$, in which

$$\tilde{\phi} = \bigodot_{1 \leq i \leq N} \phi^{(i)}. \tag{68}$$

Table 2: **Quality of the federated approximation**. The global model's performance does not critically depend on the clients' training objective; it relies only on the goodness-of-fit of their models.

| | Grid World | | Multisets | | Sequences | |
|---|---|---|---|---|---|---|
| | $L_1 \downarrow$ | Top-800 $\uparrow$ | $L_1 \downarrow$ | Top-800 $\uparrow$ | $L_1 \downarrow$ | Top-800 $\uparrow$ |
| FC-GFlowNet (CB) | 0.038 | $-6.355$ | 0.130 | 27.422 | 0.005 | $-1.535$ |
| | $(\pm 0.016)$ | $(\pm 0.000)$ | $(\pm 0.004)$ | $(\pm 0.000)$ | $(\pm 0.002)$ | $(\pm 0.000)$ |
| FC-GFlowNets (TB) | 0.039 | $-6.355$ | 0.131 | 27.422 | 0.006 | $-1.535$ |
| | $(\pm 0.006)$ | $(\pm 0.000)$ | $(\pm 0.018)$ | $(\pm 0.000)$ | $(\pm 0.005)$ | $(\pm 0.000)$ |

Notably, the best known methods for carrying out Bayesian inference over the space of phylogenetic trees are either based on Bayesian networks (Zhang and Matsen IV, 2018) or MCMC, neither of which are amenable to data parallelization and decentralized distributional approximations. More precisely, the product of Bayesian networks may not be efficiently representable as a Bayesian network, and it is usually not possible to build a global Markov chain whose stationary distribution matches the product of the stationary distributions of local Markov chains. Moreover, any categorical variational approximation factorizable over the trees' clades would not be correctly supported on the space of complete binary trees and would lead to frequently sampled invalid graphs.

### C.3 COMPARISON OF DIFFERENT TRAINING CRITERIA

**Experimental setup.** We considered the same environments and used the same neural network architectures described in subsection C.1 to parametrize the transition policies of the GFlowNets. Importantly, the implementation of the DB constraint and of the FL-GFlowNet requires the choice of a parametrization for the state flows (Bengio et al., 2023; Pan et al., 2023a). We model them as an neural network with an architecture that essentially mirrors that of the transition policies — with the only difference being the output dimension, which we set to one. Moreover, we followed suggestions in (Pan et al., 2023a; Malkin et al., 2022) and utilized a learning rate of $3 \cdot 10^{-3}$ for all parameters of the policy networks except for the partition function's logarithm $\log Z$ composing the TB constraint, for which we used a learning rate of $1 \cdot 10^{-1}$. Noticeably, we found that this heterogeneous learning rate scheme is crucial to enable the training convergence under the TB constraint.

**Further remarks regarding Figure 6.** In Figure 6, we observed that $\mathcal{L}_{CB}$ and $\mathcal{L}_{TB}$ perform similarly in the grid world and in design of sequences domains. A reasonable explanation for this is that such criteria are identically parameterized in such domains, as $\mathcal{L}_{DB}$ reduces to $R(s')p_B(s|s')p_F(s_f|s) = R(s)p_F(s'|s)p_F(s_f|s')$ in environments where every state is terminal Deleu et al. (2022). Thus, $F$ vanishes and hence the difficult estimation of this function is avoided.

### C.4 ADDITIONAL EXPERIMENTS

**Comparison between TB and CB with different learning rates.**
Figure 8 shows that increasing the learning rate for $\log Z_{\phi_Z}$ significantly accelerates the training convergence for the TB objective. In this experiment, the learning rate for the other parameters was fixed at $10^{-3}$ — following the setup of Malkin et al. (2022, Appendix B). However, CB leads to faster convergence relatively to TB for all $lr$'s. In practice, though, note that finding an adequate learning rate for $\log Z_{\phi_Z}$ may be a very difficult and computationally exhaustive endeavor that is completely avoided by implementing the CB loss.

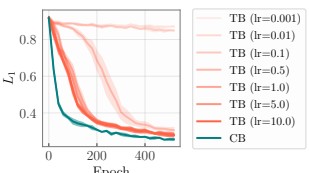

Figure 8: CB outperforms TB for different $lr$'s for $\log Z$.

**Implementing different training objectives for the clients.** Table 2 suggests that the accuracy of FC-GFlowNet's distributional approximation is mostly independent of whether the clients implemented CB or TB as training objectives. Notably, the combination phase of our algorithm is designedly agnostic to how the local models were trained — as long as they provide us with well-trained backward and forward policies. This is not constraining, however, since any practically useful training scheme for GFlowNets is explicitly based upon the estimation of such policies Malkin et al. (2022); Pan et al. (2023a); Bengio et al. (2023); Zhang et al. (2023a).

Reviewer: Xj4h

# D  RELATED WORK

**GFlowNets** were originally proposed as a reinforcement learning algorithm tailored to the search of diverse and highly valuable states within a given discrete environment (Bengio et al., 2021). Recently, these algorithms were successfully applied to the discovery of biological sequences (Jain et al., 2022), robust scheduling of operations in computation graphs (Zhang et al., 2023a), Bayesian structure learning and causal discovery (Deleu et al., 2022; 2023; da Silva et al., 2023; Atanackovic et al., 2023), combinatorial optimization (Zhang et al., 2023b), active learning (Hernandez-Garcia et al., 2023), multi-objective optimization (Jain et al., 2023), and discrete probabilistic modeling (Zhang et al., 2022a; Hu et al., 2023; Zhang et al., 2022b). Bengio et al. (2023) formulated the theoretical foundations of GFlowNets. Correlatively, (Lahlou et al., 2023) laid out the theory of GFlowNets defined on environments with a non-countable state space. Pan et al. (2023c) and Zhang et al. (2023c) extended GFlowNets to environments with stochastic transitions and rewards. Concomitantly to these advances, there is a growing literature that aims to better understand and improve this class of algorithms (Deleu and Bengio, 2023; Shen et al., 2023; Malkin et al., 2023), with an emphasis on the development of effective objectives and parametrizations to accelerate training convergence (Pan et al., 2023b;a; Malkin et al., 2022; Deleu et al., 2022). Notably, both Malkin et al. (2023) and (Zhang et al., 2023a) proposed using the variance of the a TB-based estimate of the log partition function as a training objective based on the variance reduction method of Richter et al. (2020). It is important to note one may use stochastic rewards (see Bengio et al., 2023; Zhang et al., 2023c) carry out federated inference, in the same fashion of, e.g., distributed stochastic-gradient MCMC (El Mekkaoui et al., 2021; Vono et al., 2022). Notably, stochastic rewards have also been used in the context of causal structure learning by (Deleu et al., 2022) and (Deleu et al., 2023). However, it would require many communication steps between clients and server to achieve convergence — which is precisely the bottleneck FC-GFlowNets aim to avoid.

> **Reviewer:** Xj4h

**Distributed Bayesian inference** mainly concerns the task of approximating or sampling from a posterior distribution given that data shards are spread across different machines. This comprises both federated scenarios (El Mekkaoui et al., 2021; Vono et al., 2022) or the ones in which we arbitrarily split data to speed up inference (Scott et al., 2016). Within this realm, there is a notable family of algorithms under the label of *embarrassingly parallel MCMC* (Neiswanger et al., 2014), which employ a divide-and-conquer strategy to assess the posterior. These methods sample from subposteriors (defined on each user's data) in parallel, subsequently sending results to the server for aggregation. The usual approach is to use local samples to approximate the subposteriors with some tractable form and then aggregate the approximations in a product. In this line, works vary mostly in the approximations employed. For instance, Mesquita et al. (2019) apply normalizing flows, (Nemeth and Sherlock, 2018) model the subposteriors using Gaussian processes, and (Wang et al., 2015) use hyper-histograms. It is important to note, however, that these works are mostly geared towards posteriors over continuous random variables.

**Federated learning** was originally motivated by the need to train machine learning models on privacy-sensitive data scattered across multiple mobile devices — linked by an unreliable communication network (McMahan et al., 2017). While we are the first tackling FL of GFlowNets, there are works on learning other generative models in federated/distributed settings, such as for generative adversarial networks (Hong et al., 2021; Chang et al., 2020; Qu et al., 2020) and variational autoencoders (Polato, 2021).

