# OpenReview forum: "Federated contrastive GFlowNets"
_ICLR.cc/2024/Conference — Submitted to ICLR 2024_

### Official Review · Reviewer_rsSg · 2023-10-29

**Soundness:** 2 fair
**Presentation:** 2 fair
**Contribution:** 3 good
**Rating:** 3
**Confidence:** 3

**Summary:**

This submission is mathematically exciting but, unfortunately, poorly explained and poorly motivated. The methodological section requires a much better explanation. It is currently mainly an enumeration of results. In addition, no natural problems that in any apparent way benefit from the approach are presented. Moreover, the plots in the experimental section need to be more convincing. Running time and other gains need to be made crystal clear. If this is an uncut diamond, please cut it.

**Strengths:**

This submission is mathematically exciting.

**Weaknesses:**

This submission is poorly explained and poorly motivated.

**Questions:**

"The main purpose of our experiments is to verify the empirical performance of FC-GFlowNets, i.e., their capacity to accurately sample from the combination of local rewards "
This is not sufficient.

" This is also reflected in the average reward over the top 800 samples — identical to the centralized version for FC-GFlowNet, but an order of magnitude smaller for PCVI. Again, these results corroborate our theoretical claims about the correctness of our scheme for combining GFlowNets. "
Is the centralized using a more standard GFN training approach?

" The five left- most plots indicate that the local GFlowNets were accurately trained "
Why

" Bayesian phylogenetic inference: learned × ground truth distributions. Following the pattern in Figures 2-4, the goodness-of-fit from local GFlowNets (Clients 1-5) is directly reflected in the distribution learned by FC-GFlowNet. "
Why is this good?

" In particular, GFlowNets trained via CB demonstrated faster convergence compared to current art when intermediate states are not terminal, while being com- petitive in other scenarios "
When are intermediate states terminal. Where is this shown?

---

> ### Author Response · Authors · 2023-11-14
>
> Thank you for the review. If you need any further clarification, please let us know. If you are happy with the answers, please consider raising your score.
>
> **Weaknesses**
>
> > **This submission is poorly explained and poorly motivated.**
>
> We are glad you found our work exciting and agree our motivation deserved improvement. In fact, this was also pointed out by other reviewers --- see answers to reviewers wWFS and Npeg for a thorough discussion wrt motivation. Based on that, we have expanded the 3rd paragraph in the introduction. We have also linked Appendix D in the main paper which, e.g., provides more details on federated Bayesian inference. We hope this helps elevate our work in your opinion.
>
> **Questions**
>
> > **"The main purpose of our experiments is to verify the empirical performance of FC-GFlowNets, i.e., their capacity to accurately sample from the combination of local rewards " This is not sufficient.**
>
> The overall goal of most federated learning (FL) works is to get models that preserve the same performance as a model trained in centralized settings, even if under severe communication constraints --- see, e.g., the seminal work by McMahan et al. [1]. If the FL algorithm is after point estimates, it is natural to compare federated point estimates with the one obtained with centralized data. If we are sampling (e.g., from some posterior), the analogous is comparing how well federated sampling approximates the centralized posterior [2,3].
>
> [1] Communication-Efficient Learning of Deep Networks from Decentralized Data, AISTATS 2016
>
> [2] Federated stochastic gradient Langevin Dynamics, UAI 2021
>
> [3] QLSD: Quantised Langevin Stochastic Dynamics for Bayesian Federated Learning, AISTATS 2022
>
>
>
> > **" This is also reflected in the average reward over the top 800 samples — identical to the centralized version for FC-GFlowNet, but an order of magnitude smaller for PCVI. Again, these results corroborate our theoretical claims about the correctness of our scheme for combining GFlowNets. " Is the centralized using a more standard GFN training approach?**
>
> The centralized GFlowNet is a standard GFlowNet trained to sample from $\prod_{n=1}^N R_n$. We train it using the CB loss.
>
> > **"The five left- most plots indicate that the local GFlowNets were accurately trained" why**
>
> Each plot compares the distribution of a locally-trained GFlowNet against the rewards they were trained to sample proportionally from. More specifically, we sample from the trained GFlowNet to build an explicit empirical probability mass function. Then, we compare it against the (target) probability defined by the local reward. The points in the plot show the empirical probabilities in the horizontal axis and the target one in the vertical one. Each point corresponds to some element $x \in \mathcal{X}$ in the support of the local reward.  If the empirical and target probabilities coincide, the plot should coincide on the highlighted diagonal, as is the case in these plots.
>
> > **" Bayesian phylogenetic inference: learned × ground truth distributions. Following the pattern in Figures 2-4, the goodness-of-fit from local GFlowNets (Clients 1-5) is directly reflected in the distribution learned by FC-GFlowNet. " Why is this good?**
>
> This is just a throwback to Theorem 2: good local approximations are necessary for a good global approximation. In other words, for FC-GflowNet to sample accurately from $\prod_n R_n$, each client $n$ must be well trained to sample proportional to its local reward $R_n$.
>
>
> > **" In particular, GFlowNets trained via CB demonstrated faster convergence compared to current art when intermediate states are not terminal, while being competitive in other scenarios " When are intermediate states terminal. Where is this shown?**
>
> Intermediate states are terminal whenever they belong to the support of the reward, but can also be incremented to create another valid object. For instance, in the sequence experiments, any sequence of size 5 belongs to the set of terminal states $\mathcal{X}$ but can be increased with an extra element to yield another sequence (of size 6) --- that also belongs to $\mathcal{X}$. We have included more details in Appendix C.

---

> > ### Comment · Reviewer_rsSg · 2023-11-21
> > **Motivation**
> >
> > Thanks for your work. I still find the paper very interesting but poorly motivated. I had hoped that you would provide an intersting real world application where the FC-GFlowNets approach is crucial.

---

> ### Author Response · Authors · 2023-11-21
>
> As mentioned by reviewer Xj4h, FC-GFlowNets may have a range of different applications. For instance, in drug discovery [e.g., 1-4], we can use FC-GFlowNets to scalarize different utilities/objectives or combine pre-trained models. In the context of causal discovery [e.g., 5-9], we can use FC-GFlowNets to learn cause/effect structures from federated data.  FC-GFlowNets can also scale up Bayesian inference when the likelihood is hard to evaluate, so we deliberately distribute data for computational efficiency, similar to what is done in [10-13].
>
> Would discussing these examples further in the main text alleviate your concerns? We'll be happy to incorporate suggestions.
>
> [1] Compositional Sculpting of Iterative Generative Processes. NeurIPS 2023
>
> [2] Multi-Objective GFlowNets. ArXiV 2022
>
> [3] MARS: Markov Molecular Sampling for Multi-objective Drug Discovery ICLR 2021.
>
> [4] Computer-aided multi-objective optimization in small molecule discovery. Patterns, 2023.
>
> [5] DynGFN: Bayesian Dynamic Causal Discovery using Generative Flow Networks. NeurIPS 2023.
>
> [6] Bayesian Structure Learning with Generative Flow Networks. UAI 2022
>
> [7] Joint Bayesian Inference of Graphical Structure and Parameters with a Single Generative Flow Network. NeurIPS 2023
>
> [8] Human-in-the-Loop Causal Discovery under Latent Confounding using Ancestral GFlowNets. ArXiV 2023
>
> [9] Bayesian learning of Causal Structure and Mechanisms with GFlowNets and Variational Bayes. ArXiv 2022
>
> [10] Asymptotically Exact, Embarrassingly Parallel MCMC. UAI 2014
>
> [11] Merging MCMC Subposteriors through Gaussian-Process Approximations. Bayesian Analysis, 2019.
>
> [12] Speeding Up MCMC by Efficient Data Subsampling. Journal of the American Statistical Association, 2019.
>
> [13] DG-LMC: A Turn-key and Scalable Synchronous Distributed MCMC Algorithm via Langevin Monte Carlo within Gibbs. ICML 2021.

---

### Official Review · Reviewer_Xj4h · 2023-10-30

**Soundness:** 3 good
**Presentation:** 4 excellent
**Contribution:** 3 good
**Rating:** 6
**Confidence:** 5

**Summary:**

An algorithm is proposed for training a generative flow network (GFlowNet) to match a product of distributions, each of which is sampled by a "client" GFlowNet. A training objective is stated, its correctness is proved, and bounds relating error of clients to that of the centralized model are derived. Experiments are done on federated versions of several existing tasks and on a new domain for GFlowNets (a toy case of Bayesian phylogenetic inference), where it performs well compared to a non-GFlowNet baseline.

**Strengths:**

- Excellent exposition. I have no complaints on the presentation or on the math. I easily grasped the main idea on the first reading.
  - But see one of the weaknesses on why two of the proofs could be one-line reductions to existing results.
- Beautiful and original idea. Sampling from a product of GFlowNets is natural, but the application to federated learning is creative.
  - A reference to consider adding is "Compositional sculpting of iterative generative processes" [Garipov et al., NeurIPS 2023, to appear], which considers a different kind of modular combination of samplers and could probably be used in a federated setting as well. (I am well aware that its omission is not a weakness as this paper appeared on arXiv on 28 September!)
- Application to phylogenetic inference is also a new contribution that could be expanded; I wonder how it would scale.

**Weaknesses:**

- Critical missing details for phylogenetic inference. What are the states? What are the actions?
- It should be noted that an alternative approach to the problem in the Bayesian posterior setting is to train a single GFlowNet with a *stochastic* reward, as done in [Deleu et al., UAI 2022] and [Deleu et al., NeurIPS 2023].
- The experimental validation is a little thin (esp. given the next point).
  - This is not a major weakness for me given that there may be no natural baselines, and it is made up for by the good idea and diversity of experiments.
  - However, there are still comparisons to be made, such as using different training objectives for the client models (TB or CB).
- The "contrastive loss" is claimed as original, but in fact it is not. I suggest that the authors revise the discussion on this in section 3.3 and in the claimed contributions.
  - It is well known that the expected square difference between two independent samples from a distribution is the same as twice the variance. So the loss in section 3.3 is equivalently optimizing variance of $\log P_F(\tau)-\log R(x)-\log P_B(\tau\mid x)$.
  - This variance loss is described in "GFlowNets and VI" [Malkin et al., ICLR 2023] ("local baseline"). It was independently discovered and tested in "Robust scheduling" [Zhang et al., ICLR 2023]. Such a gradient variance reduction method is originally proposed in "VarGrad" [Richter et al., NeurIPS 2020].
  - This leads me to wonder whether differences between CL and TB (Figure 6) are only due to insufficiently high learning rate on logZ for TB, or differently tuned learning rates for the policies in both algorithms.
    - Figure 7, which is hidden in the Appendix, shows that on other domains, CB performs similarly to TB/DB.
    - It seems a little misleading not to point to Figure 7 in the main text. In my opinion, it would not make the paper weaker to show all four plots in the main text and to say that first steps are made towards understanding when CB is helpful and when it is not.
  - Related to this, the proofs of all the results in section 3.3 have one-line reductions to existing results (the TB training theorem and the TB gradient analysis theorem). The current proofs are quite obfuscated.
- The bound in Theorem 2 is nice but may not be very useful in practice. If one of the models is missing a mode, which is quite possible with reverse-KL objectives like those used here, $\alpha_n$ will be very small, and the final bound is not useful. Therefore, I wonder if one can obtain similar bounds on other divergences that are less sensitive than the Jeffrey divergence to mode-dropping.

I am happy to increase the score from 5 to 6 or even 8 if these are satisfactorily addressed.

**Questions:**

I like the motivation paragraph on the first page, but what do you see as the main future applications: Privacy-preserving distributed training? Large-scale Bayesian inference where the full reward is expensive to compute on a single client? Modular combination of pretrained GFlowNets?

In particular, I am not sure to understand the private-rewards application. If each GFlowNet fits (close to) perfectly and can share its policies, then each client's reward density can be recovered as the ratio of its forward and backward policies.

---

> ### Author Response · Authors · 2023-11-14
> **Answer to Reviewer Xj4h, Part 1/2**
>
> We appreciate your thoughtful feedback. Hopefully, the answers below sufficiently address your concerns. Please let us know if something else needs further clarification or if we missed something.
>
> **Weaknesses**
>
> > **Critical missing details for phylogenetic inference. What are the states? What are the actions?**
>
> **A:** In the phylogeny experiments, a state is represented as a forest. Initially, each leaf belongs to a different singleton tree. An action consists of picking two trees and joining their roots to a newly added node. The generative process is finished when all nodes are connected in a single tree. We understand that this description was missing in the document, and we included it in Appendix C.1, along with an illustrative picture.
>
> > **It should be noted that an alternative approach to the problem in the Bayesian posterior setting is to train a single GFlowNet with a stochastic reward, as done in [Deleu et al., UAI 2022] and [Deleu et al., NeurIPS 2023]**
>
> **A:** Thanks for pointing this out. We are unsure about the relevance of the two references to stochastic rewards — both focus on causal structure learning. If they were not a typo, could you elaborate on that?
>
> Nonetheless, we could use Zhang et al.'s (ICML 2023) scheme for training GFlowNets in environments with stochastic rewards drawn from the distribution
>
> $$
> R(x) = R_{c}(x), \, c \sim q(\cdot | \{1, \dots, K\}), % \{R_{1}(x), \dots, R_{K}(x)\}),  .
> $$
>
> in which $q$ is a categorical distribution over the clients $\{1, \dots, K\}$. By assigning a weight $q(c)^{-1}$ to the $c$th client's reward, $R_{c}(x)^{q(c)^{-1}}$, Proposition 1 of Zhang et al. ensures that the trained GFlowNet would sample an object $x$ with probability proportional to
>
> $\exp \underset{c \sim q}{\mathbb{E}} \left[\log R\_{c}(x)^{q(c)^{-1}}\right]  = \underset{1 \le k \le K}{\prod} {R}_{{k}}(x) $
>
> The downside is that this requires many communication steps between clients and server. This is precisely the bottleneck we are trying to avoid with FC-GFlowNets, imposing one *single communication step* between the clients and the server. We wrote a short version of this discussion in related works (Appendix D).
>
>
>
> > **However, there are still comparisons to be made, such as using different training objectives for the client models (TB or CB).**
>
> **A:** Thanks for the suggestion. In principle, FC-GFlowNets are agnostic to how the local models were trained as long as they provide us with forward and backward policies. We have now included Table 2 (below) in Appendix C, comparing the performance of FC-GFlowNets when clients are trained with TB vs. CB. The metrics are all very similar.
>
> |                          		 | Grid World         		 |       		 | Multisets         		 |      		 | Sequence          		 |       		 |
> |-----------------------------------|-------------------------|-------------------------|-------------------------|-------------------------|-------------------------|--------------------|
> |                          		 | L₁ ↓          		 | Top-800 ↑      		 | L₁ ↓          		 | Top-800 ↑      		 | L₁ ↓          		 | Top-800 ↑      		 |
> | FC-GFlowNet (CB)         		 | 0.038          		 | -6.355         		 | 0.130          		 | 27.422         		 | 0.005          		 | -1.535         		 |
> |                          		 | (± 0.016)      		 | (± 0.000)      		 | (± 0.004)      		 | (± 0.000)      		 | (± 0.002)      		 | (± 0.000)      		 |
> | FC-GFlowNets (TB)        		 | 0.039          		 | -6.355         		 | 0.131          		 | 27.422         		 | 0.006          		 | -1.535         		 |
> |                          		 | (± 0.006)      		 | (± 0.000)      		 | (± 0.018)      		 | (± 0.000)      		 | (± 0.005)      		 | (± 0.000)      		 |
>
>
>
> > **The "contrastive loss" is claimed as original, but in fact it is not. I suggest that the authors revise the discussion on this in section 3.3 and in the claimed contributions**
>
> **A:**  We once again thank you for noticing the relationship between the contrastive loss (CL) and the log-partition variance loss (VL) of Zhang et al. (ICLR 2023) — which we did not notice.  In fact, CL equals twice VL *in expectation*. Nonetheless, as you pointed out, CL and VL use different estimators for the variance up to some positive multiplicative constant. Additionally, our balance condition (CB) is novel per se and essential to deriving the federated loss (Corollary 1), which is the basis of our method.  We have updated the introduction, section 3.2 (which introduces the CB), and related works (Appendix D) to acknowledge this work.

---

> ### Author Response · Authors · 2023-11-14
> **Answer to Reviewer Xj4h, Part 2/2**
>
> > **This leads me to wonder whether differences between CL and TB (Figure 6) are only due to insufficiently high learning rate on logZ for TB, or differently tuned learning rates for the policies in both algorithms.**
>
> **A:** To verify whether our gains stemmed from a poor choice of learning rate, we have run additional experiments (in Figure 8), sliding it from $0.001$ to $10$. In all cases, we saw CL converging to better optima than TB. That being said, while there may be some learning rate for log Z leading to better convergence, finding it may be difficult. We have updated Figure 6 to include the results from (former) Figure 7.
>
> > **Related to this, the proofs of all the results in section 3.3 have one-line reductions to existing results (the TB training theorem and the TB gradient analysis theorem). The current proofs are quite obfuscated.**
>
> **A:** We tried to make the proofs as clear and as self-contained as possible. In particular, as our arguments in Section 3.2 consist primarily of standard and easy-to-follow calculations, we believe that articulating a comprehensive proof in the Appendix is beneficial for enhancing the results' clarity. It is unclear whether shortening the proofs would make them easier to follow.
>
> > **The bound in Theorem 2 is nice but may not be very useful in practice.**
>
> **A:** We did not intend Theorem 2 to be used for diagnosing pathological implementations of FC-GFlowNet. Theorem 2 only aims to illustrate that the global distribution may diverge considerably from the target distribution when a client's policies are inadequately trained. Particularly, if one of the clients fails to detect some modes, as you suggested, then the global distribution will also miss these modes. This effect, similarly observed in e.g. De Souza et al. (AISTATS 2022), is accurately represented by our upper bound on the Jeffrey's divergence.
>
> We agree that developing good diagnostics for federated GFlownets is an important topic that we would like to pursue in future works. However, the design of tight divergence bounds and effective diagnostic methods is currently an unaddressed issue in the GFlowNets literature, even for centralized settings.
>
>
> **Questions**
>
> > **I like the motivation paragraph on the first page, but what do you see as the main future applications: Privacy-preserving distributed training? Large-scale Bayesian inference where the full reward is expensive to compute on a single client? Modular combination of pretrained GFlowNets?**
>
>
> **A:** Thank you for the question — those are precisely the three main applications we envision for federated GFlowNets. We realized this was hard to grasp for most reviewers and decided to elaborate further on this paragraph and the conclusion.
>
> We believe that the distributed training of GFlowNets will be invaluable in addressing large-scale Bayesian inference problems, e.g., structure learning (Deleu et al., NeurIPS 2023) and phylogenetic inference (Altekar et al., Bioinformatics 2004), in which the reward function may be expensive to evaluate due to the size of the conditioning dataset.
>
> > **In particular, I am not sure to understand the private-rewards application. If each GFlowNet fits (close to) perfectly and can share its policies, then each client's reward density can be recovered as the ratio of its forward and backward policies.**
>
> **A:** Indeed, the rewards can be retrieved if the GFlowNets are perfectly trained.  However, having access to the reward does not entail, e.g., disclosing the data used to compute a likelihood (or a local posterior up to some multiplicative constant). In this case, the local GFlowNet can be seen as a proxy that provides some notion of privacy w.r.t. local data.

---

> ### Comment · Reviewer_Xj4h · 2023-11-18
> **Thank you for the response**
>
> Thanks for the clarifications and new experiments, which have improved the paper. Just to clarify a couple more things:
>
> > We tried to make the proofs as clear and as self-contained as possible...
>
> I don't think repeating a difficult proof is worth it when there is such a simple reduction (zero CL implies TB satisfied for some value of $\log Z$). I may be unable to convince you on this, but at least I think that **the existence of such a reduction should be mentioned** (could be just a sentence in the main text saying that CL implies TB for the obvious reason) -- it would only make the paper stronger!
>
> > We are unsure about the relevance of the two references to stochastic rewards — both focus on causal structure learning. If they were not a typo, could you elaborate on that?
>
> No, not a typo. In those papers, the goal is to sample a posterior $p(G\mid D)$ where $D=\{x_i\}_{i=1}^n$. The reward is the joint $p(G,D)=p(G)\prod_ip(x_i\mid G)$. To avoid scoring all samples $x_i$ using the latent structure graph $G$, a stochastic reward of $p(G)p(x_i)^n$ for a random $x_i$ is used. This gives an unbiased estimate of the log-reward and therefore of the loss. Such an approach is applicable to the general setting where the reward decomposes as prior times product of data-dependent terms, including the phylogenetic tree setting you consider.
>
> > Nonetheless, we could use Zhang et al.'s (ICML 2023) scheme for training GFlowNets in environments with stochastic rewards
>
> I am not sure which paper you are talking about here.

---

> > ### Author Response · Authors · 2023-11-18
> >
> > Thanks for engaging and for the suggestions to improve our work.
> >
> > > I don't think repeating a difficult proof is worth it when there is such a simple reduction...
> >
> > Regarding the proofs of Lemma 1 and Theorem 3, we have included short reductions to the TB results in the main paper (pg 6). Please let us know if this is what you had in mind.
> >
> > > No, not a typo. In those papers, the goal is to sample a posterior...
> >
> > We were unaware that these works on GFlowNets for causal discovery used stochastic rewards. Thanks for pointing this out! They’re now cited in related works (pg. 21). Regarding “Zhang et al. (ICML 2023)”, we were the ones committing a typo. We meant this work: https://arxiv.org/abs/2302.05793 — also cited in the same paragraph of pg 21.

---

> > > ### Comment · Reviewer_Xj4h · 2023-11-18
> > > **Response**
> > >
> > > Thanks for clarifying. That paper is not published in ICML 2023, and the proposition there is not an original result (actually, the fact is already in "GFlowNet foundations", section 3.3.2).

---

> > > > ### Author Response · Authors · 2023-11-19
> > > >
> > > > Thanks for the input. We have updated our related works section to acknowledge this earlier work.

---

### Official Review · Reviewer_Npeg · 2023-10-31

**Soundness:** 2 fair
**Presentation:** 3 good
**Contribution:** 2 fair
**Rating:** 5
**Confidence:** 3

**Summary:**

The paper studies the problem of training a GFlowNet in federated manner. Each client has its own reward function and local data. And each client trains its local GFlowNet. The goal here for the server is learn a GFlowNet model that samples data proportional to the product of clients reward. The paper conducts experiments on various tasks.

**Strengths:**

1. The paper studies the unexplored problem of federated training of GFlowNet.
2. The paper conducts extensive experiments on various tasks.

**Weaknesses:**

1. I believe should give more concrete example on why learning a GFlowNet samples data proportional to product of reward is important in federated setting. From reading the paper and experiments, I am not clear about the significance of the problem studied by this paper. In fact, it seems the problem tackled by this paper is an special case while it is not evident that this special case is important in federated setting.
2. It seems that the main contribution of this paper is its problem setting where learning a GFlowNet model that samples data proportional to product of rewards can be done in a federated fashion. However, given the problem, learning the global model aggregating local GFlowNet seems to be straightforward. Therefore, I think I believe the contribution of the paper in federated learning is not high.

**Questions:**

Please see weaknesses.

---

> ### Author Response · Authors · 2023-11-14
>
> Dear reviewer, thank you for your feedback. We realize the motivation provided in the introduction was brief, which may have underplayed the impact of our contribution. We hope our answers clear your concerns, and we have incorporated their content in the revised manuscript. If your concerns are sufficiently solved, please consider raising your score. Otherwise, we will be happy to engage further and provide more clarifications.
>
>
> > **Q: I am not clear about the significance of the problem studied by this paper. In fact, it seems the problem tackled by this paper is an special case while it is not evident that this special case is important in federated setting.**
>
> **A:** The most common problem in federated learning is learning some point estimate $\theta^{\star}$ that minimizes the sum of loss functions  $\ell_{1}, \ldots, \ell_K$ for each client $k=1\ldots K$. Typically, the loss is equivalent to the negative log-likelihood of $\theta$ on some local dataset $\mathcal{D}_k$. In the Bayesian realm, the corresponding problem is sampling from a posterior distribution $p(\theta | \mathcal{D})$ proportional to the products of the local likelihoods and some prior over theta. Each of these terms in the product can be seen as a local reward. Then, the product of local rewards is proportional to the posterior. That being said, FC-GFlowNets are a natural solution for Bayesian federated learning cases in which $\theta$ is a discrete-valued random variable.  More generally, FC-GFlowNets work for any reward that factorizes as products. Thus, it can be used for, e.g., multi-objective active learning. Importantly, there well-established interest of the ML community in Bayesian federated learning — see., e.g., Appendix D for more details. To clarify the significance of our problem, we have also extended paragraph 3 (Introduction) in the revised manuscript. Please see our first answer to reviewer wWFS for further discussion about the motivation.
>
> > **Q: It seems that the main contribution of this paper is its problem setting where learning a GFlowNet model that samples data proportional to product of rewards can be done in a federated fashion. However, given the problem, learning the global model aggregating local GFlowNet seems to be straightforward.**
>
> **A:** We are not aware of any simpler approach to sampling from the product of GFlownets. The naive strategy would be to train a GFlowNet to sample from the product of terminal distributions $\hat{R} = \prod_{k=1}^{K} p_k^\intercal(x)$ of the clients $k=1\ldots K$. Note, however, that $p_k^\intercal(x)$ requires summing over all terminal paths leading to $x$ — which is infeasible in practice. Furthermore, using a Monte Carlo estimate of $p_k^\intercal(x)$ plus some standard GFlowNet loss results in biased gradients, which are undesirable if we want to converge with stochastic gradient descent.
>
> Furthermore, sampling from products of arbitrary distributions over discrete objects is not a trivial task. This is the reason, for instance, why most of the literature on parallel MCMC focuses solely on continuous cases. Importantly, while there is a good body of literature on Bayesian federated learning, to the best of our knowledge, FC-GFlowNets is the first method capable of handling discrete cases.

---

> > ### Comment · Reviewer_Npeg · 2023-11-23
> >
> > I appreciate the authors response. However, the rebuttal does not address most of my concerns. I still believe that the motivation of this work need some improvements.
> >
> > First, it seems that the paper formalizes the objective function of training a GFlowNet model as the product of local objectives for training a GFlowNet locally. GFlowNet is a generative model and as a result there might be a user that wants to use GFlowNet to generate new samples with desirable characteristics with respect to their own objective. I do not understand the motivation behind learning a global GFlowNet while clients have different objectives. I understand that federated learning can be helpful in cases where clients do not have enough data to train a good model locally. However, when it comes to training a GFlowNet with limited data another alternative is to leverage active learning techniques where the model is trained iteratively and at each iteration generated samples are added to the initial dataset. My main point is federated training of a GFlowNet is not well-motivated when clients have different objectives and data distributions. The author response did not address this concern.
> >
> > Another thing that prevents me to vote positively for this paper is that I think the problem studied by this paper can be solved by applying existing Bayesian federated learning methods. Then the paper only introduces an unexplored problem in federated learning and the paper does not overcome any considerable challenge in federated learning. Moreover, the paper does not show that the problem is important by giving concrete examples in real world applications. The response does not indicate the importance of the studied problem. This limits the contribution of the paper from federated learning point of view,
> >
> > To sum up, reading the rebuttal I decide to stay with my initial score. I am sorry if I get back to authors late. However, authors are welcome to respond to my comments again and I will definitely consider the authors response carefully.

---

> ### Author Response · Authors · 2023-11-23
> **Thanks for engaging**
>
> Thanks for engaging in the discussion. To allow further interaction, we've crafted a very to-the-point answer --- touching key points of your reply. Please let us know if this clarifies your concerns. If not, we hope this message opens a channel for further clarification.
>
> > I do not understand the motivation behind learning a global GFlowNet while clients have different objectives.
>
> **Local posteriors are realistically never the same.** Note that the GFlowNet’s training objective (reward function) is dependent on the clients’ data. For example, in a Bayesian federated learning setting, each client possesses a different data set and, in particular, has a different training objective (posterior distribution) upon which to train a GFlowNet. Thus, except in the very unlikely scenario in which the posterior distribution is the same for all clients, the objective function will differ among the clients.
>
> **Our method also applies to multi-objective optimization.** We also note that the utility of our setup goes beyond federated Bayes. As we have discussed in our answer to reviewer wWFS, our proposal can be applied to multi-objective optimization, where the goal is to obtain samples with high values across a pool of utility functions. For instance, in drug discovery, a key challenge is finding molecules that satisfy multiple constraints (e.g., affinity, solubility, safety).  In a setting where different experts work independently to obtain property-specific models using GFlowNets, this challenge can be cast as sampling from the product of these local GFlowNets. Note also that our work allows for reusing pre-trained GFlownNets to sample from a weighted product, effectively reusing these models as pointed by reviewer X4jh.
>
>
> > However, when it comes to training a GFlowNet with limited data another alternative is to leverage active learning techniques where the model is trained iteratively and at each iteration generated samples are added to the initial dataset.
>
> **GFlowNets are not used for data augmentation.** We would like to emphasize that GFlowNet is not a generative model designed to mimic the (unknown) data distribution and subsequently generate more samples; instead, it is a method tailored to sampling from an unnormalized distribution over compositional objects like graphs. In this sense, GFlowNets are one-of-a-kind, different from VAEs, GANs, Normalizing Flows, etc. In fact, GFlowNets are closer to Markov Chain Monte Carlo (MCMC) than to other deep generative models.
>
>
> > I think the problem studied by this paper can be solved by applying existing Bayesian federated learning methods…
>
> **Existing federated sampling method cannot handle discrete-valued random variables.** We would like to clarify that this problem (federated Bayes on discrete posteriors) cannot be solved using previous techniques from the Bayesian federated learning literature.  By discrete posteriors, we mean that we are sampling from a discrete-valued random variable with a reward proportional to said posterior. The vast majority of the literature on Bayesian federated learning (as known as federated sampling) relies on either gradients wrt the parameter of interest (e.g., SGLD-based methods) . The works on embarrassingly parallel Bayes also do not apply, since they impose continuous approximations (e.g., Gaussians and KDEs) to the local contributions/subposteriors
>
> ---
> We do hope for, and would greatly appreciate, your stronger support for this work once you feel your concerns have been addressed. Otherwise, please do not hesitate to engage further.

---

> > ### Comment · Reviewer_Npeg · 2023-11-23
> >
> > Thanks for your response. I agree with you that GFlowNet does not mimic the unknown data distribution to generate new samples. However, I do not agree with the argument that since GFlowNet does not mimic the unknown data distribution, it cannot be used for data augmentation. In contrast, I believe data augmentation can improve GFlowNet training. Data augmentation for generative methods such as diffusion models and VAE should be performed carefully to preserve the original data distribution since these methods work based on mapping the data distribution to a noise. However, GFlowNet is trained using labeled data and learns a policy that samples objects proportional to their reward regardless of labeled data distribution. As the number of labeled data increases the performance of GFlowNet may improve. Therefore, one can add new data samples to the available labeled data without being concerned about the data distribution. Also, the paper "Biological Sequence Design with GFlowNets" published by ICML 2022 trains the GFlowNet by adding generated data step by step to the initial data. The experiments in the mentioned work show that the resulting GFlowNet model works better than GFlowNet without data augmentation. Furthermore, reading the paper and rebuttal it is not obvious to me that what is the technical and theoretical challenges of handling discrete-valued random variables.

---

> > > ### Author Response · Authors · 2023-11-23
> > >
> > > > However, GFlowNet is trained using labeled data and learns a policy that samples objects proportional to their reward regardless of labeled data distribution. As the number of labeled data increases the performance of GFlowNet may improve. Therefore, one can add new data samples to the available labeled data without being concerned about the data distribution.  Also, the paper "Biological Sequence Design with GFlowNets" published by ICML 2022 trains the GFlowNet by adding generated data step by step to the initial data. The experiments in the mentioned work show that the resulting GFlowNet model works better than GFlowNet without data augmentation.
> > >
> > > We believe there is a misunderstanding here. In the paper you shared, GFlowNet is used to sample from an acquisition function $F$ inside an active learning loop. Effectively, the acquisition depends on the model $M$, which gets updated with new data. Say $M_i$ is the predictive model at the start of round $i$, and the respective acquisition (which is also the reward) is $F(M_i)$. The GFlowNet is simply trained to sample proportional to the acquisition. Note that the GFlowNet, however, needs to sample proportional to $F$. It does not make sense to say the GFlowNet improves between acquisition rounds; the GFlowNet either samples from $F(M_i)$ correctly or it does not. If $M^{\star}$ is the perfect model and, in iteration $i=1$, the GFlowNet samples from $F(M^{\star})$ instead of $F(M_1)$, the GFlowNet is incorrectly trained.
> > >
> > > In Bayesian terms, suppose the GFlowNet is trained to sample from a posterior $p(\theta|\mathcal{D})$ which has mass $0.6$ on the "true" value of $\theta$. However, for some reason, it learns to sample $\theta$ with probability $0.99$. If this happens, the GFlowNet is poorly trained since it is not sampling proportional to the posterior.
> > >
> > > > Furthermore, reading the paper and rebuttal it is not obvious to me that what is the technical and theoretical challenges of handling discrete-valued random variables.
> > >
> > > Gradient-based sampling cannot be used here, which renders most existing federated sampling methods useless. This means you would need gradients of the reward $R : \mathcal{X} \to \mathbb{R}$ with respect to inputs $x \in \mathcal{X}$ but $\mathcal{X}$ is not continuous --- there are no gradients by definition.  If you see, e.g., [1, 2], you'll notice gradients of the log prior + log-likelihood (i.e., the reward function) throughout.
> > >
> > > [1] https://proceedings.mlr.press/v161/mekkaoui21a/mekkaoui21a.pdf
> > >
> > > [2] https://proceedings.mlr.press/v151/vono22a/vono22a.pdf

---

> ### Comment · Reviewer_Npeg · 2023-11-23
>
> Thank you for your response. I believe there is not any misunderstanding here. I am confident that experiments in the mentioned paper show that GFlowNet-AL show better performance than GFlowNet in some biological sequence design tasks.

---

> > ### Author Response · Authors · 2023-11-23
> >
> > Thanks for the quick reply and for the chance to elaborate on the difference in the evaluation setup between the work you cited and ours.
> >
> > Please note that performance in the work you cited is not a measure of how well we sample from $R$. For instance, in that work, 'performance' is the mean score for a sample of biological sequences. In our work, performance means quantifying how well the model samples from the distribution defined by the (normalized) reward. Furthermore, the samples from GFLowNet-AL are not from a single GFlowNet --- as shown in Algorithm 1, at each round they train a different GFlowNet and store samples from it.
> >
> > Thanks again for your review and for the engagement.

---

### Official Review · Reviewer_wWFS · 2023-10-31

**Soundness:** 4 excellent
**Presentation:** 4 excellent
**Contribution:** 3 good
**Rating:** 6
**Confidence:** 3

**Summary:**

This paper proposes a novel framework for federated learning of GFlowNets, called FC-GFlowNet, which is based on a divide-and-conquer strategy that requires only a single communication step. The authors introduce a new concept, contrastive balance, which provides necessary and sufficient conditions for the correctness of general GFlowNets. The paper also presents experiments demonstrating the effectiveness of FC-GFlowNets in various settings, including grid-world, sequence, and multiset generation, and Bayesian phylogenetic inference. Overall, the paper provides a significant contribution to the community interested in generative modeling, enabling GFlowNets in federated settings, introducing a novel training scheme for centralized settings, and demonstrating the potential of GFlowNets on a new application domain.

**Strengths:**

1. The paper provides a contribution to the community interested in generative modeling, enabling GFlowNets in federated settings, introducing a novel training scheme for centralized settings, and demonstrating the potential of GFlowNets on a new application domain.
  2. The authors introduce a new concept, contrastive balance, which provides necessary and sufficient conditions for the correctness of general GFlowNets, and demonstrate its effectiveness in training GFlowNets.

**Weaknesses:**

1. However, the motivation for introducing GFlowNets into Federated Learning (FL) appears to be lacking. Despite the existing research gap, what specific challenges or benefits does FL encounter that require the integration of GFlowNets?

2. Conversely, is there a scenario in which GFlowNets necessitate Federated Learning (FL)? In my view, the explanation provided in the third paragraph is somewhat lacking. It would be helpful to provide more practical examples to illustrate this.

3. The experimental setup regarding the clients is not sufficiently clear. Are all clients identical, or does this setup involve non-iid (non-independent and identically distributed) problems?

4. It would enhance clarity if the authors could include a figure or pseudocode illustrating the algorithm.

**Questions:**

See Weaknesses

---

> ### Author Response · Authors · 2023-11-14
>
> Thank you for your feedback. We hope the answers below solve your issues. If your concerns were not cleared, please let us know and we'll try to clarify them further. Otherwise, please consider raising your score.
>
>
> >**Q:** However, **the motivation for introducing GFlowNets into Federated Learning (FL) appears to be lacking**. Despite the existing research gap, **what specific challenges or benefits does FL encounter that require the integration of GFlowNets? Conversely, is there a scenario in which GFlowNets necessitate Federated Learning (FL)?**
>
>
> **A:** Thanks for the opportunity to motivate Federated GFlowNets further. As mentioned in the Introduction, we foresee at least two impactful applications of FC-GFlowNets (our proposal): multi-objective active learning and Federated Bayesian Inference.
>
> In multi-objective active learning, we want to obtain samples with high values across a pool of utility functions. For instance, in drug discovery, a key challenge is finding molecules that satisfy multiple constraints (e.g., affinity, solubility, safety). In a setting where different experts work independently to obtain property-specific models using GFlowNets, this challenge can be cast as sampling from the product of these local GFlowNets. This corresponds to the scalarization approach for multi-objective optimization, but we are interested in sampling to ensure diversity instead of maximizing --- which was also one of the primary motivations behind GFlowNets.
>
> Federated Bayesian Inference/Learning entails sampling from a posterior distribution $p(\theta|\mathcal{D})$ where $\mathcal{D}$ is a dataset partitioned into a number of clients $k=1\ldots K$, each with a subset $\mathcal{D}_k$. The common setting is that the posterior factorizes as
>
> $p(\theta|\mathcal{D}) \propto p(\theta) \prod_{k} p(\mathcal{D}_k | \theta)$, where $p(\theta)$ is a prior and $p(\mathcal{D}_k | \theta)$ is the likelihood contribution of client $k$. Note that the likelihood is typically different for each $k$; more generally, they don't even need to share a functional form. There are a series of methods to solve this problem in a single communication round when $\theta$ is a real-valued random variable. FC-GFlowNets (our proposal) solves this problem for the discrete case — and is the first work to do so.
>
> Thanks to your comment, we have extended paragraph 3 (Introduction) in the revised manuscript. We also highlighted that our related work section discusses the Federated Bayes setting in detail (Appendix D).
>
> > **Q: The experimental setup regarding the clients is not sufficiently clear. Are all clients identical, or does this setup involve non-iid (non-independent and identically distributed) problems?**
>
> **A:** In our experiments, clients are never identical, i.e., there is no pair of local rewards $(R_i, R_j)$ that are equal. For instance, in the phylogeny experiments, each client's reward depends on different parts of the nucleotide sequences. In this case, no two clients have access to the same part of the sequence. For the grid world, each client has different reward beacons. Finally, the multiset and design-of-sequences experiment, clients value items differently. We have clarified that no clients have identical rewards in the experiment section.
>
>
> > **Q: It would enhance clarity if the authors could include a figure or pseudocode illustrating the algorithm.**
>
> **A:** Thanks for the suggestion. We have included a pseudo-code for FC-GFlowNets in the Appendix.

---

> > ### Comment · Reviewer_wWFS · 2023-11-22
> >
> > My concerns have been well addressed by the response of the authors, I would raise my score.

---

### Author Response · Authors · 2023-11-20
**Author/Reviewer Discussion**

Dear Reviewers,

We understand that reviewing is time- and effort-consuming, and we are grateful for your service.
We've carefully answered your questions and updated the manuscript accordingly.

As we approach the end of the discussion period, we would greatly value your feedback on whether we have effectively addressed your concerns. To simplify tracking changes in the manuscript, we have colored all changed/new text excerpts blue. We have also marked them with reviewer IDs, depending on whose concerns are being addressed.

Thank you very much for your time and dedication to this process. We look forward to your valuable feedback and continuing our dialogue.

Best regards,
The Authors

---

### Author Response · Authors · 2023-11-23
**Summary**

We are grateful to all the reviewers for their time and insightful comments, as well as to the (senior) area, program, and general chairs for their service to the community.

We are glad that the reviewers note that our work is the first one to apply GFlowNets to federated learning and distributed inference (X4jh, wWFS, Npeg), demonstrates the potential of GFlowNets in the new domain of Bayesian phylogenetic Inference (X4jh, wWFS), and conducts extensive experiments in various tasks (Npeg). Also, our work is clearly presented (X4jh, wWFS) and exciting (rsSg).

To the best of our efforts, we've tried to address all the specific comments, including the minor ones, that have been raised by each reviewer. In particular, we have:
1. rewritten parts of the introduction to clarify our work’s motivation, adding further examples regarding Bayesian federated learning and drug discovery (Section 1)
2. updated the proofs to elucidate how some of our theoretical results can be derived from previously published ones (Lemma 1 and Theorem 3’s proofs)
3. updated the related works section to acknowledge the use of stochastic rewards as an alternative to dealing with expensive-to-evaluate reward functions in GFlowNet training (Appendix D)
4. clarified the versatility of our method, including that it applies to multiobjective optimization and combining pre-trained GFlowNets (Section 1)
5. included a pseudocode to shed light on the training algorithm of FC-GFlowNets (Algorithm 1)
6. executed additional experiments to compare the goodness-of-fit of the global model when using different training objectives for the clients (Table 2)
7. improved our conclusion to elucidate the future applications of FC-GFlowNets, namely, large-scale Bayesian inference and modular combination of pretrained GFlowNets (Section 5)
8. elucidated the description of the generative process of phylogenetic trees (Figure 7)
9. compared the proposed contrastive balance loss with the trajectory balance loss implemented with different learning rates for the parameter corresponding to the log-partition function (Figure 8)

We believe that acting on reviewers’ feedback has reinforced the strengths of our work, and we thank them again for their very constructive comments.

---

### Meta-Review · Area_Chair_oDaR · 2023-12-06

**Metareview:**

This paper introduces a novel framework, FC-GFlowNet, for federated learning of GFlowNets, employing a divide-and-conquer strategy that remarkably requires only a single communication step. The authors introduce the concept of contrastive balance, establishing necessary and sufficient conditions for the correctness of general GFlowNets. The paper substantiates its claims with experiments showcasing the effectiveness of FC-GFlowNets across diverse settings, including grid-world, sequence, multiset generation, and Bayesian phylogenetic inference. Overall, the paper makes a significant contribution to the generative modeling community by enabling GFlowNets in federated settings, presenting a novel training scheme for centralized settings, and demonstrating the potential of GFlowNets in a new application domain.

However, the motivation for introducing GFlowNets into Federated Learning (FL) lacks clarity, and the benefits that FL brings to GFlowNets remain unclear. Additionally, the experimental setup concerning the clients is not sufficiently elucidated. Following extensive discussion, all reviewers do not express strong support for this paper.

**Justification For Why Not Higher Score:**

N/A

**Justification For Why Not Lower Score:**

N/A

---

### Decision · Program_Chairs · 2024-01-16

Reject